# Cost-aware Stopping for Bayesian Optimization

Qian Xie [* 1]  Linda Cai [* 2]  Alexander Terenin [1]  Peter I. Frazier [1]  Ziv Scully [1]

## Abstract

In automated machine learning, scientific discovery, and other applications of Bayesian optimization, deciding when to stop evaluating expensive black-box functions in a cost-aware manner is an important but underexplored practical consideration. A natural performance metric for this purpose is the *cost-adjusted simple regret*, which explicitly captures the trade-off between solution quality and cumulative evaluation cost. Existing stopping rules for Bayesian optimization are either heuristic, or are theoretically grounded but designed to optimize simple regret without accounting for evaluation costs; as a result, they provide no guarantees against unnecessary evaluations when costs are high. We propose a *principled cost-aware stopping rule* for Bayesian optimization that adapts to varying evaluation costs without heuristic tuning. Our rule is grounded in a theoretical connection to state-of-the-art cost-aware acquisition functions, namely the Pandora's Box Gittins Index (PBGI) and log expected improvement per cost (LogEIPC). When paired with either acquisition function, we prove that the resulting policy satisfies a theoretical guarantee bounding the expected cost-adjusted simple regret. Across synthetic tasks and empirical benchmarks including hyperparameter optimization and neural architecture size search, pairing our stopping rule with PBGI or LogEIPC usually matches or outperforms other acquisition-function–stopping-rule pairs in terms of cost-adjusted simple regret. Code available at: HTTPS://GITHUB.COM/QIANJANEXIE/COSTAWARESTOPPINGBAYESOPT.

*Equal contribution [1]Cornell University, Ithaca, New York, USA [2]University of California, Berkeley, Berkeley, California, USA. Correspondence to: Qian Xie <qx66@cornell.edu.>.

*Proceedings of the 43rd International Conference on Machine Learning*, Seoul, South Korea. PMLR 306, 2026. Copyright 2026 by the author(s).

## 1. Introduction

Bayesian optimization is a framework designed to efficiently find approximate solutions to optimization problems involving expensive-to-evaluate black-box functions, where derivatives are unavailable. Such problems arise in applications like hyperparameter tuning (Snoek et al., 2012), robot control optimization (Martinez-Cantin, 2017), and material design (Zhang et al., 2020). It works by iteratively, (a) forming a probabilistic model of the black-box objective function based on data collected thus far, then (b) optimizing an acquisition function, which balances exploration-exploitation tradeoffs, to carefully choose a new point at which to observe the unknown function in the next iteration.

In this work, we consider the *cost-aware* setting, where one must pay a cost to collect each data point, and study *adaptive stopping rules* that choose when to stop the optimization process. After stopping at some terminal time, we measure performance in terms of simple regret, which is the difference in value between the best solution found so far and the global optimum. Collecting a data point can reduce simple regret, but incurs cost in order to do so.

As an example, consider using a cloud computing environment to tune the hyperparameters of a classifier in order to optimize a performance metric on a given test set. Training and evaluating test error takes some number of CPU or GPU hours, that may depend on the hyperparamaters used. These come with a financial cost, billed by the cloud computing provider, which define our cost function. The objective value is the business value of deploying the trained model under the given hyperparameters—a given function of the model's accuracy. From this perspective, simple regret can be understood as the opportunity cost for deploying a suboptimal model instead of the optimal one. Motivated by the need to balance cost-performance tradeoff in examples such as above, we aim to design stopping rules that optimizes *expected cost-adjusted simple regret*, defined as the sum of simple regret and the cumulative cost of data collection.

Several stopping rules have been proposed for Bayesian optimization. Simple heuristics—such as fixing the number of iterations or stopping when the best value remains unchanged for a certain number of iterations—are widely used, but can either stop too early or lead to unnecessary evaluations. Other approaches include acquisition-function-based

rules that stop when metrics such as probability of improvement (Lorenz et al., 2015) fall below a preset threshold, or when expected improvement (EI) (Nguyen et al., 2017; Zhou et al., 2024) or knowledge gradient (KG) (Frazier & Powell, 2008) fall below the cost per sample. Regret-bound-based rules instead aim to optimize simple regret (Makarova et al., 2022; Ishibashi et al., 2023; Wilson, 2024; Li et al., 2023). These approaches are primarily developed for the classical uniform-cost setting and do not explicitly account for varying evaluation costs. Our work instead provides a principled stopping rule with theoretical guarantees that applies both to the uniform-cost setting and more generally to varying evaluation costs.

In this work, we study how to design cost-aware stopping rules, motivated by two primary factors. First, state-of-the-art cost-aware acquisition functions such as the Pandora's Box Gittins Index (PBGI) (Xie et al., 2024) and log expected improvement per cost (LogEIPC) (Ament et al., 2023) have not yet been studied in the adaptive stopping setting. As our experiments in Section 4 show, introducing adaptive stopping can further improve the performance of these acquisition functions compared to fixed-budget evaluations. Second, while regret-based stopping rules such as UCB–LCB (Makarova et al., 2022) guarantee low simple regret, they often incur excessive evaluation costs, resulting in high cost-adjusted regret, as reflected in our experiments.

Our work builds upon the Bayesian-optimal Pandora's Box decision principle underlying the PBGI acquisition function and extends it to the adaptive stopping setting in correlated Bayesian optimization. Furthermore, we show that an existing stopping rule proposed for the EI acquisition function in the classical cost-unaware setting (Nguyen et al., 2017; Zhou et al., 2024), admits a principled choice of the stopping threshold as the evaluation cost per sample, rather than a heuristic tuning parameter. Under this choice, the resulting stopping condition is equivalent to the PBGI stopping rule, and naturally extends to the cost-aware setting. This stopping rule can therefore naturally be paired with acquisition functions derived from either the PBGI or the EI design principles. Acquisition functions such as KG or MES, which are derived from alternative principles such as value-of-information or entropy search, inherently require their own matched stopping rules.

Our contributions are summarized as follows:

1. *A Novel Cost-Aware Stopping Rule.* We present an adaptive stopping rule derived from Pandora's box theory, which also naturally extends to the expected improvement design principle, establishing a unified and principled framework applicable to both classic and cost-aware Bayesian optimization.

2. *Theoretical Guarantees.* We prove in Theorem 3.2 that

our stopping rule, when paired with the PBGI or LogEIPC acquisition function, satisfies a theoretical upper bound on the expected cost-adjusted regret, which constitutes the first theoretical guarantee of this type for any adaptive stopping rule for Bayesian optimization.

3. *Empirical Validation.* We conduct a systematic empirical evaluation across multiple acquisition-function—stopping-rule pairs in cost-aware Bayesian optimization. Our results on synthetic tasks and empirical benchmarks show that pairing our proposed stopping rule with the PBGI or LogEIPC acquisition function usually matches or outperforms other pairs.

## 2. Bayesian Optimization and Adaptive Stopping

In black-box optimization, the goal is to find an approximate optimum of an unknown objective function $f : X \to \mathbb{R}$ using a limited number of function evaluations at points $x_1, \ldots, x_T \in X$ where $X$ is the search space and $T$ is a given search budget, potentially chosen adaptively. The convention measures performance in terms of *expected simple regret* (Garnett, 2023, Sec. 10.1), given by

$$\mathcal{R} = \mathbb{E}\left[ \min_{1 \le t \le T} f(x_t) - \inf_{x \in X} f(x) \right], \qquad (1)$$

where the expectation is taken over all sources of randomness, including the sequence of points $x_1, \ldots, x_T$ selected by the algorithm. Bayesian optimization approaches this problem by building a probabilistic model of $f$—typically a Gaussian process (GP) (Rasmussen & Williams, 2006), conditioned on the observed data points $(x_t, y_t)_{t=1}^T$, where $y_t = f(x_t)$. For each iteration $t = 1, \ldots, T$, an acquisition function $\alpha_t : X \to \mathbb{R}$ then guides the selection of new samples by carefully balancing the exploration-exploitation tradeoff arising from uncertainty about $f$.

### 2.1. Cost-aware Bayesian Optimization

*Cost-aware Bayesian optimization* (Lee et al., 2020; Astudillo et al., 2021) extends the above setup to account for the fact that evaluation costs can vary across the search space. For instance, in the hyperparameter tuning example of Section 1, costs can vary according to the time needed to train a machine learning model under a given set of hyperparameters $x$. Cost-aware Bayesian optimization handles this by introducing a cost function $c : X \to \mathbb{R}^+$, which may be known or unknown ahead. The cost function, or observed costs, are then used to construct the acquisition function $\alpha_t$.

In this work, we focus primarily on the *cost-per-sample* formulation (Chick & Frazier, 2012; Cashore et al., 2016; Xie et al., 2024) of cost-aware Bayesian optimization, which seeks methods with stopping time $\tau$, not necessarily fixed,

that achieve a low *expected cost-adjusted simple regret*

$$\mathcal{R}_c = \mathbb{E}\left[ \underbrace{\min_{1 \le t \le \tau} f(x_t) - \inf_{x \in X} f(x)}_{\text{simple regret}} + \underbrace{\sum_{t=1}^{\tau} c(x_t)}_{\text{cumulative cost}} \right]. \quad (2)$$

One can also work with the *expected budget-constrained* formulation (Xie et al., 2024), which incorporates budget constraints explicitly, and seeks algorithms which achieve a low expected simple regret under an expected evaluation budget. Here, performance is evaluated in terms of

$$\mathcal{R} = \mathbb{E}\left[ \min_{1 \le t \le \tau} f(x_t) - \inf_{x \in X} f(x) \right]$$

$$\text{where } x_1, \ldots, x_\tau \text{ satisfy} \quad \mathbb{E}\sum_{t=1}^{\tau} c(x_t) \le B. \quad (3)$$

The stopping rules we study can be applied in this setting as well: we discuss this in Section 3.1.

For both settings, we work with a few acquisition functions—chiefly, the *Pandora's Box Gittins Index (PBGI)* (Xie et al., 2024) and *log expected improvement per cost (LogEIPC)* (Xie et al., 2024) (the log variant (Ament et al., 2023) of the expected improvement per cost (Snoek et al., 2012)). At iteration $t$, let $x_{1:t} := \{x_1, \ldots, x_t\}$ be the evaluated points, $y_{1:t} := \{y_1, \ldots, y_t\}$ be their observed values, $y_{1:t}^* = \min_{1 \le s \le t} y_s$ be the current best observed value, and $f \mid x_{1:t}, y_{1:t}$ be the posterior distribution of $f$. Then the PBGI acquisition function is defined as

$$\alpha_t^{\mathrm{PBGI}}(x) :=$$
$$\begin{cases} g & \text{s.t. } \mathrm{EI}_{f|x_{1:t}, y_{1:t}}(x; g) = c(x), & x \notin x_{1:t}, \\ f(x), & x \in x_{1:t}, \end{cases} \quad (4)$$

where the value $g$ is the threshold such that the expected improvement over $g$ equals the cost[1] and $\mathrm{EI}_\psi(x; y) = \mathbb{E}\left[\max(y - \psi(x), 0)\right]$ is the expected improvement at point $x$ with respect to some random function $\psi : X \to \mathbb{R}$ and a baseline value $y$. The LogEIPC acquisition function is defined as

$$\alpha_t^{\mathrm{LogEIPC}}(x) := \log \frac{\mathrm{EI}_{f|x_{1:t}, y_{1:t}}(x; y_{1:t}^*)}{c(x)}, \quad (5)$$

i.e., the logarithm of the expected improvement divided by the cost. We also consider classical cost-unaware acquisition functions—*lower confidence bound (LCB)* and *Thompson sampling (TS)*; see Garnett (2023) for details.

---

[1] For an evaluated point $x$, we set $g = f(x)$ since the posterior at $x$ collapses to a point mass at the observed value $f(x)$ and its evaluation cost can be treated as 0.

## 2.2. Adaptive Stopping Rules

To the best of our knowledge, stopping rules for Bayesian optimization typically do not incorporate cost explicitly into the stopping criterion. We broadly categorize existing methods as follows:

**Criteria based on convergence or significance of improvement.** This includes empirical convergence, namely stopping when the best value remains unchanged for a fixed number of iterations, or the *global stopping strategy (GSS)* (Bakshy et al., 2018), which stops when the improvement is no longer statistically significant relative to the inter-quartile range (i.e., the range between the 25th percentile and the 75th percentile) of prior observations. In the multi-fidelity setting, Foumani & Bostanabad (2025) proposed stopping when the high-fidelity surrogate's estimated optimum has stabilized over a window of iterations, which is related to cost-awareness but differs from our setting with explicitly varying evaluation costs.

**Acquisition-based criteria.** This includes stopping rules based on acquisition functions such as PI, EI, and KG. These approaches typically stop when the acquisition value falls below a threshold—such as a predetermined constant (Lorenz et al., 2015; Locatelli, 1997), the median of the initial acquisition values (Ishibashi et al., 2023), or the cost of sampling (Nguyen et al., 2017; Zhou et al., 2024; Frazier & Powell, 2008). They are typically defined via fixed thresholds and are primarily designed for the uniform-cost setting, without explicitly accounting for varying evaluation costs.

**Regret-based criteria.** This includes stopping rules based on confidence bounds such as *UCB-LCB* (Makarova et al., 2022) and the *gap of expected minimum simple regrets* (Ishibashi et al., 2023). These stop when certain estimated regret bounds fall below a preset or data-driven threshold. The related *probabilistic regret bound (PRB)* stopping rule (Wilson, 2024) stops when estimated simple regret is below a small threshold $\epsilon$ with confidence $1 - \delta$. In contrast to these forward-looking rules, Li et al. (2023) propose a backward-looking stopping rule that detects when the search has entered a locally convex region and stops once an estimate of local regret falls below a predetermined threshold.

In settings beyond Bayesian optimization, Chick & Frazier (2012) have proposed a cost-aware stopping rules for finite-domain sequential sampling problems with independent values. Although this formulation allows for varying costs, it does not extend to general Bayesian optimization settings which use correlated GP models.

# 3. A Stopping Rule Based on the Pandora's Box Gittins Index

In this work, we propose a new stopping rule tailored for two state-of-the-art acquisition functions used in cost-aware Bayesian optimization: PBGI and LogEIPC, introduced in Section 2. As discussed by Xie et al. (2024), these acquisition functions are closely connected, arising from two different approximations of the intractable dynamic program which defines the Bayesian-optimal policy for cost-aware Bayesian optimization. We now show that this connection can be used to obtain a principled stopping criterion to be paired with both acquisition functions.

**A stopping rule for PBGI.** To derive a stopping rule for PBGI, consider first the Pandora's Box problem, from which it is derived. Pandora's Box can be seen as a special case of cost-per-sample Bayesian optimization, as introduced in Section 2, where $X = \{1, \ldots, N\}$ is a discrete space and $f$ is assumed uncorrelated. In this setting, the observed values do not affect the posterior distribution—meaning, we have $f(x') \mid x_{1:t}, y_{1:t} = f(x')$ at all unobserved points $x'$—and thus the acquisition function value at point $x$ is time-invariant and can be written simply as $\alpha^{\mathrm{PBGI}}(x)$, where $\mathrm{EI}_f(x; \alpha^{\mathrm{PBGI}}(x)) = c(x)$. Following Weitzman (1979), one can show using a Gittins index argument that selecting $x_t$ to minimize $\alpha^{\mathrm{PBGI}}$ is Bayesian-optimal under minimization—the algorithm which does so achieves the smallest expected cost-adjusted regret, among all adaptive algorithms.

One critical detail applied in the optimality argument of Weitzman (1979) is that the policy defined by $\alpha^{\mathrm{PBGI}}$ is *not* Bayesian-optimal for any fixed deterministic $T$. Instead, it is optimal only when the stopping time $T$ is chosen according to the condition

$$\min_{x \in X \setminus \{x_1, \ldots, x_t\}} \alpha^{\mathrm{PBGI}}(x) \geq y_{1:t}^*, \tag{6}$$

where $\geq$ can be replaced by $>$[2], and as before, $y_{1:t}^*$ is the best value observed so far.

In the correlated setting we study, Xie et al. (2024) extend $\alpha^{\mathrm{PBGI}}$ to define an acquisition function by recomputing it at each iteration $t$ based on the posterior mean and variance, which defines $\alpha_t^{\mathrm{PBGI}}$ as in Equation (4). In order to also extend the associated stopping rule, a subtle design choice arises: should we use $\alpha_{t-1}^{\mathrm{PBGI}}$ (before posterior update) or $\alpha_t^{\mathrm{PBGI}}$ (after posterior update) in Equation (6)? While prior theoretical work (Gergatsouli & Tzamos, 2023) adopts the former choice, we instead use the latter (after posterior

update), resulting in the stopping rule

$$\min_{x \in X \setminus \{x_1, \ldots, x_t\}} \alpha_t^{\mathrm{PBGI}}(x) \geq y_{1:t}^*. \tag{7}$$

This choice is motivated by two reasons. First, we argue it more faithfully reflects the intuition behind Weitzman's original stopping rule. In the independent setting, $\alpha^{\mathrm{PBGI}}(x)$ represents a kind of *fair value* for point $x$ (Kleinberg et al., 2016). For an evaluated point $x \in \{x_1, \ldots, x_t\}$, this is simply the observed function value. For an unevaluated point $x \notin \{x_1, \ldots, x_t\}$, the fair value reflects uncertainty in $f(x)$ and the cost $c(x)$. The stopping rule Equation (7) then says to *stop when the best fair value is among the already-evaluated points*—namely, when no point provides positive expected gain relative to its cost if evaluated in the next round, conditioned on current observations. In the correlated setting, the fair value naturally extends to $\alpha_t^{\mathrm{PBGI}}$, because this incorporates all known information at a given time. Second, we show in Appendix C.2 that using $\alpha_t^{\mathrm{PBGI}}$ yields tangible empirical gains in cost-adjusted regret.

**Connection with LogEIPC.** The above reasoning may at first seem to be fundamentally tied to the Pandora's Box problem and its Gittins-index-theoretic properties. We now show that it admits a second interpretation in terms of log expected improvement per cost, which arises from a completely different one-time-step approximation to the Bayesian-optimal dynamic program for cost-aware Bayesian optimization in the general correlated setting.

To show this, we start with definitions above and apply a sequence of transformations. Recall that for any unevaluated point $x \notin \{x_1, \ldots, x_t\}$, $\alpha_t^{\mathrm{PBGI}}(x)$ is defined in Equation (4) to be the solution to

$$\mathrm{EI}_{f \mid x_{1:t}, y_{1:t}}(x; \alpha_t^{\mathrm{PBGI}}(x)) = c(x), \tag{8}$$

and since $\mathrm{EI}_\psi(x; y)$ is strictly increasing in $y$, any inequality involving $\alpha_t^{\mathrm{PBGI}}(x; c)$ can be lifted through the EI function without changing its direction, which means $\alpha_t^{\mathrm{PBGI}}(x) \geq y_{1:t}^*$ holds if and only if $\mathrm{EI}_{f \mid x_{1:t}, y_{1:t}}(x; y_{1:t}^*) \leq c(x)$. This implies that the PBGI stopping rule from Equation (7) is equivalent to stopping when

$$\mathrm{EI}_{f \mid x_{1:t}, y_{1:t}}(x; y_{1:t}^*) \leq c(x), \quad \forall x \in X \setminus \{x_1, .., x_t\}, \tag{9}$$

meaning *stop when no point's expected improvement is worth its evaluation cost*. Rearranging the inequality and taking logs, we can rewrite this condition using the LogEIPC acquisition function[3]:

$$\max_{x \in X \setminus \{x_1, \ldots, x_t\}} \alpha_t^{\mathrm{LogEIPC}}(x; y_{1:t}^*) \leq 0. \tag{10}$$

---

[2]Following Xie et al. (2024, Appendix B), optimality holds under any tie-breaking rule in the cost-per-sample setting, but only under a carefully-chosen stochastic tie-breaking rule in the expected budget-constrained setting. For simplicity, we use the stopping-as-early-as-possible tie-breaking rule in this paper.

[3]Excluding evaluated points is not theoretically required but often beneficial in practice, as numerical stability adjustments can cause their expected improvement to remain positive.

When $c(x) \equiv c_0$, Equation (8) reduces to the EI thresholding stopping rule (Nguyen et al., 2017; Zhou et al., 2024)

$$\max_{x \in X \setminus \{x_1, \ldots, x_t\}} \alpha_t^{\text{EI}}(x; y_{1:t}^*) \leq c_0. \quad (11)$$

We call the stopping rule given by the equivalent conditions (Equations (7), (9) and (10)) the *PBGI/LogEIPC stopping rule*. Figure 1 gives an illustration of how the rule behaves in a simple setting, demonstrating that it is more conservative—preferring to stop earlier—when the cost is high.

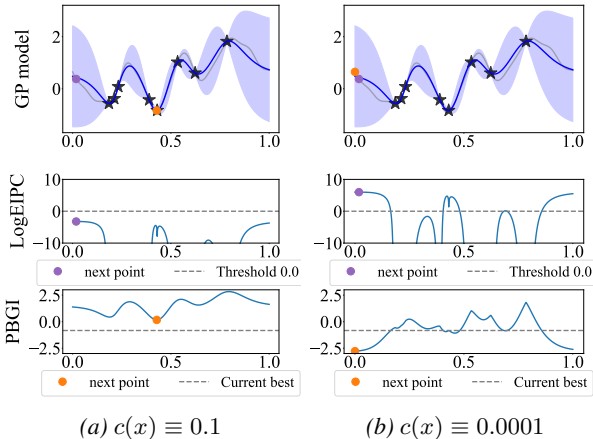

*(a) $c(x) \equiv 0.1$    (b) $c(x) \equiv 0.0001$*

*Figure 1.* Illustration of the PBGI/LogEIPC stopping rule under a uniform-cost setting. When the cost-per-sample is large ($c(x) \equiv 0.1$), the minimum PBGI acquisition value exceeds the threshold (current best observed value) and the maximum LogEIPC acquisition value falls below the threshold 0.0, indicating stopping; when the cost-per-sample is small ($c(x) \equiv 0.0001$), the minimum PBGI acquisition value is smaller than the threshold (current best observed value) and the maximum LogEIPC acquisition value remains above the threshold 0.0, indicating no stopping.

### 3.1. Theoretical Guarantees on Cost-Adjusted Regret

The connection between PBGI and LogEIPC in fact goes beyond a shared stopping rule. In Lemma 3.1, we prove that when paired with this stopping rule, both acquisition functions guarantee that at each iteration before stopping, the expected improvement at the selected point is at least its evaluation cost.

**Lemma 3.1** (Expected improvement exceeds cost before stopping). *Let $X$ be compact, and let $f : X \to \mathbb{R}$ be a random function with prior mean $\mu(\cdot)$. Consider a Bayesian optimization algorithm that begins at some initial point $x_1 \in X$ with cost $C = c(x_1)$, acquires subsequent points using either the PBGI or LogEIPC acquisition function, and terminates according to the PBGI/LogEIPC stopping rule. Let $\tau = \min_{t \geq 1}\{\sup_{x \in X} \alpha_t^{\text{LogEIPC}}(x) \leq 0\}$ be the algorithm's stopping time, and denote the posterior expected improvement function by $\alpha_t^{\text{EI}}(x) = \text{EI}_{f|x_{1:t}, y_{1:t}}(x; y_{1:t}^*)$. Then, for all $t < \tau$, $\alpha_t^{\text{EI}}(x_{t+1}) \geq c(x_{t+1})$.*

As a consequence of this claim, we show in Theorem 3.2 that the PBGI/LogEIPC stopping rule, when paired with PBGI or LogEIPC, achieves cost-adjusted simple regret no worse than stopping immediately after the initial evaluation. Notably, many acquisition function–stopping rule pairs fail to satisfy this *no-worse-than-immediate* property in practice (see Figures 2 and 3), largely because most stopping rules are designed with simple regret alone in mind and do not explicitly account for evaluation costs. Moreover, this guarantee is tight in the worst case: when evaluation costs are sufficiently large relative to the maximum achievable improvement, immediate stopping is optimal (see Proposition A.3). To our knowledge, this is the first formal guarantee on cost-adjusted simple regret for Bayesian optimization, generalizing beyond simpler settings such as Pandora's Box, which assumes independent and discrete evaluations.

**Theorem 3.2** (No worse than immediate stopping). *Consider the setting and algorithm specified in Lemma 3.1. Let $U := \mu(x_1) - \mathbb{E}[\min_{x \in X} f(x)] < \infty$, then the algorithm's expected cost-adjusted simple regret is bounded by*

$$\mathbb{E}\left[ y_{1:\tau}^* - \min_{x \in X} f(x) + \sum_{t=1}^{\tau} c(x_t) \right]$$
$$\leq \mathbb{E}[\underbrace{y_1 - \min_{x \in X} f(x) + c(x_1)}_{\text{cost-adjusted regret of immediate stopping}}] = U + C. \quad (12)$$

This result yields a bound on expected cumulative cost and, under the natural assumption that practical evaluations have a positive minimum cost, a high-probability finite-time termination guarantee.

**Corollary 3.3** (Finite-time termination under positive costs). *Consider the setting and algorithm specified in Lemma 3.1. Then the expected cumulative cost of the algorithm is bounded by $\mathbb{E}\left[\sum_{t=1}^{\tau} c(x_t)\right] \leq U + C$. Further, if evaluation costs are uniformly bounded below by a constant $c_0 > 0$, i.e., $c(x) \geq c_0, \forall x \in X$, then for any $\delta \in (0,1)$, the algorithm terminates in at most $\frac{U+C}{\delta \cdot c_0}$ iterations with probability $1 - \delta$.*

*Remark* 3.4. Lemma 3.1 and Theorem 3.2 are stated for simplicity under initialization from a single point $x_1$. The results extend directly to arbitrary initial datasets $(x_{1:t_0}, y_{1:t_0})$ by conditioning on the observations and treating the resulting posterior as the effective prior. The same guarantees then hold for subsequent iterations $t \geq t_0$, with all quantities defined with respect to this updated posterior.

#### 3.1.1. IMPLICATION IN THE BUDGETED SETTING

We first note that in the discrete Pandora's Box setting, under an expected budget constraint $B$, minimizing $\alpha_t^{\text{PBGI}}(x)$ is Bayesian-optimal: it achieves the lowest *expected simple regret, among all algorithms* satisfying the expected cumulative cost within the budget (i.e., $\mathbb{E}\left[\sum_{t=1}^{\tau} c(x_t)\right] \leq B$),

and the cost function used in defining $\alpha_t^{\mathrm{PBGI}}(x)$ is $\lambda c(x)$ for some *cost-scaling factor* $\lambda$ which depends on $B$ (Xie et al., 2024, Theorem 2).

This result, at first, appear to be completely Pandora's-Box-theoretic: it requires $X$ to be discrete and $f$ to be independent. In the more general correlated setting of cost-aware Bayesian optimization, however, Bayesian optimality of PBGI may no longer hold, and the relationship between $B$ and the choice of $\lambda$ is not immediately clear. Theorem 3.2 helps bridge this gap: it provides an upper bound on the expected cumulative cost up to the stopping time, which in turn yields the following principled choice of $\lambda$ that ensures compliance with the budget constraint.

**Corollary 3.5** (Choosing $\lambda$ for expected budget compliance)**.** *Consider the setting, algorithm, and notation specified in Lemma 3.1 and Theorem 3.2, but with costs rescaled by a factor $\lambda > 0$: both the acquisition values and the stopping conditions are computed using $\lambda c(\cdot)$. For some budget $B > C$, if the cost-scaling factor is set to $\lambda = \frac{U}{B-C}$, then the algorithm's expected cumulative unscaled cost satisfies $\mathbb{E}\left[\sum_{t=1}^{\tau} c(x_t)\right] \leq B$.*

If this choice of $\lambda$ proves overly conservative—leading to underspending within the budget—it can be paired with the PBGI-D variant of Xie et al. (2024), which starts with $\lambda = \lambda_0$ and halves it each time stopping is triggered. Choosing $\lambda_0 = U/(B - C)$ aligns the initial fixed-$\lambda$ phase with the budget in expectation, while ensuring that the adaptive decay is activated as designed. In Figure 10 in Appendix C, we show PBGI-D with this choice of $\lambda_0$ is competitive.

All proofs in this section are deferred to Appendix A.

### 3.2. Practical Implementation Considerations for the PBGI/LogEIPC Stopping Rule

**Expressing objective and cost in common units.** Evaluation costs are often measured in different units than the objective, such as time versus accuracy. To compare them directly, we rescale costs by a constant $\lambda > 0$, the conversion rate between objective improvement and cost. For instance, if one is willing to spend 1000 seconds to improve test accuracy by 0.01, then $\lambda = 10^{-5}$. Importantly, in the cost-per-sample setting we mainly study, $\lambda$ is a fixed constant set by the problem provider, rather than a parameter tuned heuristically by the user. The budgeted setting without a natural unit conversion is discussed in Section 3.1.1.

**Unknown costs.** In practice, evaluation costs are often not known in advance. They can be modeled either (i) deterministically, using domain knowledge (e.g., proportional to model size in hyperparameter tuning), or (ii) stochastically, via assumptions on the distribution of $c(x)$ (our theoretical guarantees still hold after replacing $c(x)$ with $\mathbb{E}[c(x)]$). In

Section 4, we present empirical results under both modeling approaches. For the latter, we follow Astudillo et al. (2021, Proposition 2) to model $\ln c(x)$ as a GP with posterior mean $\mu_{\ln c}$ and variance $\sigma_{\ln c}^2$. To compute PBGI or LogEIPC, as well as their related stopping rules (including those in the prior work and ours), we replace $c(x)$ in Equations (4), (5) and (9) with $\mathbb{E}[c(x)] = \exp(\mu_{\ln c}(x) + (\sigma_{\ln c}(x))^2/2)$. The empirical performance is preserved. See Appendix C.3 for further discussion.

**Preventing spurious stops.** Although our stopping rule has theoretical guarantees, in practice, it—like other adaptive stopping rules—can still suffer from spurious stops caused by two main sources. First, early in the optimization process, the GP model parameters often fluctuate significantly as new data points are collected, causing unstable stopping signals. To mitigate this, we enforce a stabilization period consisting of the first few evaluations, during which we do not allow any stopping rule to trigger. Second, imperfect optimization of the acquisition function—which is especially common in continuous higher-dimensional search spaces—can lead to misleading stopping signals. To handle this, we smooth the stopping signal in this setting by applying a moving average over a window of $W$ iterations, and require the stopping condition to hold consistently under this smoothed signal before stopping. See Figure 4 for an illustration. We use the same window size $W = 20$ for both the stabilization period and smoothing to minimize the number of parameters in our main experiments. A ablation study on the effect of window size can be found in Appendix C.4.1.

## 4. Experiments

To evaluate our proposed PBGI/LogEIPC stopping rule, we design three complementary sets of experiments that progressively evaluate its performance. First, we consider an idealized, low-dimensional Bayesian regret setting in which the GP model exactly matches the true objective. This controlled setting allows us to isolate the effect of different stopping rules without interference from model misspecification and acquisition function optimization errors. Then, we move to a higher-dimensional Bayesian regret setting where acquisition-function optimization is imperfect. Finally, we evaluate in practical settings with objective model misspecification, using the LCBench hyperparameter tuning benchmark and the NATS neural architecture size search benchmark. In each case, we compare pairs of acquisition functions and stopping rules, which we describe next.

**Acquisition functions.** We consider four common acquisition functions that were discussed in Section 2: PBGI, LogEIPC, LCB, and TS—chosen for their competitive performance, computational efficiency, and close connections

to existing stopping rules.

**Baselines.** We compare the proposed PBGI/LogEIPC stopping rule against several stopping rules from prior work: *UCB–LCB* (Makarova et al., 2022), *LogEIPC-med* (Ishibashi et al., 2023), *SRGap-med* (Ishibashi et al., 2023), and *PRB* (Wilson, 2024). We also include two simple heuristics used in practice: *Convergence* and *GSS*. *UCB–LCB* stops once the gap between upper and lower confidence bounds falls below a configurable threshold $\theta$. *LogEIPC-med* stops when the log expected improvement per cost drops beneath $\log(\eta)$ plus the median of its initial $I$ values. *SRGap-med* stops when the gap of the expected minimum simple regret falls below $\chi$ times the median of its initial $I$ values. *PRB* triggers once a probabilistic regret bound satisfies the regret tolerance $\epsilon$ and confidence $\delta$ parameters. *Convergence* stops as soon as the best observed value remains unchanged for $k$ iterations. *GSS* stops if the recent improvement is less than $\phi \times \text{IQR}$ over the past $k$ trials where IQR denotes the inter-quartile range of current observations. Finally, we include two reference baselines: *Immediate*, which stops right after the initial evaluation (see Section 3.1), and *Hindsight*, which, for each trial, selects the stopping time that yields the lowest cost-adjusted simple regret in hindsight, thus providing a lower bound on achievable performance.

In our experiments, unless specified otherwise, we follow parameter values recommended in the literature, and set $\theta = 0.01$, $\eta = 0.01$, $\chi = 0.01$, $I = 20$, $\epsilon = 0.1$ for Bayesian regret experiments, $\epsilon = 0.5\%$ of the best test error for empirical experiments, $\delta = 0.05$, $k = 5$, and $\phi = 0.01$. For the stabilization and smoothing window, we use $W = 20$, aligned with the initial window size $I = 20$ used in EI-med (Ishibashi et al., 2023).

Each experiment is repeated with 50 random seeds to assess variability, and we report the mean with error bars (2 times the standard error) for each stopping rule. Each trial, in the sense of a run with a distinct random seed, is capped at a fixed number of iterations; if a stopping rule is not triggered within this limit, the stopping time is set to the cap. Details are provided in Appendix B.

### 4.1. Bayesian Regret

We first evaluate our PBGI/LogEIPC stopping rule on random functions sampled from prior. Specifically, objective functions are sampled from Gaussian processes with Matérn 5/2 kernels with length scale 0.1, defined over spaces of dimension $d = 1$ and $d = 8$. We consider a variety of evaluation cost function types, including uniform costs, linearly increasing costs in terms of parameter magnitude, and periodic costs that vary non-monotonically over the domain.

In the 1-dimensional experiments, we perform an exhaus-

tive grid search over $[0, 1]$ to isolate the effect of stopping behavior from numerical optimization. In the 8-dimensional setting, due to instability from higher dimensionality, we apply moving average with a window size of 20 when computing the PBGI/LogEIPC stopping condition. See Figure 4 in Appendix B for an illustration. More details on our experiment setup, and computational considerations are provided in Appendices B and C.

Figure 2 shows a comparison of cost-adjusted regret for pairings of acquisition functions and stopping rules under different cost-scaling factors $\lambda = 0.1, 0.01, 0.001$ in the 1-dimensional setting, and under different cost regimes (uniform, linear, periodic) in 8-dimensional setting.

In the 1-dimensional setting, the optimization problem is relatively straightforward and all acquisition functions have nearly identical hindsight optima, independent of cost scale or cost type. From the plots, our PBGI/LogEIPC stopping rule, when paired separately with the PBGI and LogEIPC acquisition functions, delivers competitive performance for each $\lambda$, and is particularly strong in handling high-cost scenarios—those with large $\lambda$.

In the 8-dimensional experiments, however, clear gaps emerge. Under uniform and linear costs, PBGI, LogEIPC, and LCB exhibit similar hindsight optima, substantially better than TS, while in the periodic case, PBGI and LogEIPC achieve much better hindsight optima than LCB and TS. In every cost regime, combining our stopping rule with the PBGI or LogEIPC acquisition function yields cost-adjusted regret that nearly matches the hindsight optimal.

This demonstrates that our PBGI/LogEIPC stopping rule is relatively robust to the increased optimization difficulty. Further discussions of our experiments across multiple cost types and cost-scaling factors can be found in Appendix C.

### 4.2. Automated Machine Learning Benchmarks

We then evaluate on two empirical automated machine learning benchmarks based on use cases from hyperparameter optimization and neural architecture search: LCBench (Zimmer et al., 2021) and NATS-Bench (Dong et al., 2021). Unless otherwise specified, results in this section assume a known cost function; results under the unknown-cost setting are provided in Appendix C.

LCBench (Zimmer et al., 2021) provides training data of 2,000 configurations over a 7-dimensional hyperparameter space, evaluated on 35 OpenML (Vanschoren et al., 2014) datasets. In the *unknown-cost* experiments, the cost is the full 51-epoch training time. For the known-cost setting, we observe that training time scales approximately linearly with the number of model parameters. Based on a linear regression fit (see Appendix B), we adopt $\alpha p$ as a proxy for training time, where $p$ is the number of model parameters

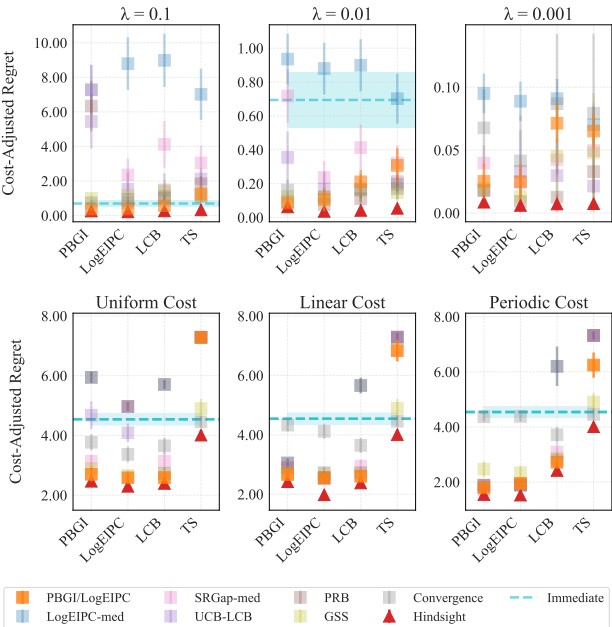

*Figure 2.* Cost-adjusted simple regret across acquisition-stopping rule pairs in 1D and 8D Bayesian regret setting. In 1D, objective functions are sampled from a GP with a Matérn-5/2 kernel and a linear cost function scaled by $\lambda = 0.1, 0.01, 0.001$. The Immediate baseline is omitted at $\lambda = 0.001$ due to its much higher regret (mean 0.6942, error bar [0.5314, 0.8570]). In 8D, objective functions are also drawn from a GP with a Matérn-5/2 kernel, using three cost functions scaled by $\lambda = 0.01$.

and $\alpha$ is a dataset-specific estimated coefficient.

NATS-Bench (Dong et al., 2021) provides a search space of 32,768 neural architectures varying in channel sizes across layers, evaluated on three datasets. As in LCBench, we similarly approximate the full 90-epoch (training and validation) runtime using a linear function $\alpha F + \beta$ of the number of floating point operations $F$; see Appendix B for details.

We report the mean and error bars (2 times standard error) of the *cost-adjusted simple regret*, where we consider the evaluation costs to be some representative cost-scaling factor $\lambda$ (see Section 3.2) multiplied with cumulative runtime. Simple regret is computed as the difference between (a) test error of the configuration with best validation error at the stopping time and (b) the best test error over all configurations. We present the results using proxy runtime here, and defer to Appendix C the additional results under (i) the *cost model misspecification* scenario (proxy runtime used during Bayesian optimization but actual runtime used for final performance evaluation), and (ii) the *unknown-cost* scenario (actual runtime used during Bayesian optimization).

Figure 3 presents performance of acquisition function–stopping rule pairs on LCBench and NATS-Bench. For LCBench, we report min–max normalized cost-adjusted simple regret (see definition in Appendix C) aggregated

across 35 datasets in Figure 3 evaluated under three representative[4] cost-scaling values. For NATS-Bench, we report cost-adjusted simple regret under $\lambda = 10^{-5}$, with additional results under two other representative cost-scaling factors shown in Figures 14 to 16 in Appendix C.

Per-dataset LCBench results are provided in Figures 19 to 21 in the appendix. Across the 35 LCBench tasks, around 75% show our PBGI/LogEIPC stopping rule performing competitively when paired with either PBGI or LogEIPC. The remaining outliers are mostly (aside from two exceptions) very small datasets with $< 10000$ instances (i.e., data points) that might lead to severe model misspecification. We also report the rank frequency of our PBGI/LogEIPC stopping rule when paired separately with the PBGI and LogEIPC acquisition functions, evaluated on medium-to-large datasets (with $> 10,000$ instances) and all datasets; see Table 1. A diagnosis of the underperforming datasets can be found in Appendix C.3.6, where we find that the apparent underperformance is likely driven by benchmark-intrinsic val/test mismatch rather than by the stopping rule itself.

Additional results under the cost model misspecification setting and the unknown-cost setting are provided in Figures 17 and 18 in Appendix C, along with comparisons between our adaptive stopping rule and fixed-iteration baselines. Overall, our PBGI/LogEIPC stopping rule consistently performs strongly in terms of cost-adjusted simple regret, when paired with either PBGI or LogEIPC. We also report, in Table 2 of Appendix C, how often each stopping rule fails to trigger within the 200-iteration cap: baselines such as SRGap-med and UCB–LCB often fail to stop early, particularly on NATS-Bench, whereas ours reliably stops before the cap.

In empirical benchmarks such as LCBench and NATS-Bench, where objective misspecification relative to the GP prior is unavoidable, our stopping rule remains robust. When paired with PBGI, it delivers performance close to the hindsight optimal, except on two NATS tasks with potential severe misspecification. Pairing with LogEIPC is slightly less competitive, likely reflecting PBGI's stronger robustness under misspecification. Indeed, Figure 10 in Appendix C compares acquisition functions under fixed-budget evaluations (without stopping) and shows that PBGI consistently attains lower simple regret than LogEIPC. Thus, while our stopping rule applies to both acquisition functions, the matched PBGI combination is a particularly strong default under objective misspecification.

---

[4]These $\lambda$ values are chosen to avoid degenerate cases—neither so large that the policy stops after only a few evaluations (e.g., $\lambda = 10^{-4}$ on NATS-Bench, see Figure 17) nor so small that evaluations are effectively free.

| $\lambda$ | Acq | Top 1 | | Top 2 | | Top 3 | |
|---|---|---|---|---|---|---|---|
| | | Large | All | Large | All | Large | All |
| $10^{-3}$ | PBGI | 20.0% | 20.0% | 70.0% | 48.6% | 80.0% | 60.0% |
| | LogEIPC | 40.0% | 31.4% | 65.0% | 45.7% | 70.0% | 62.8% |
| $10^{-4}$ | PBGI | 40.0% | 28.6% | 70.0% | 51.5% | 80.0% | 62.9% |
| | LogEIPC | 30.0% | 22.9% | 50.0% | 45.8% | 85.0% | 68.7% |
| $10^{-5}$ | PBGI | 50.0% | 28.6% | 70.0% | 51.5% | 90.0% | 74.4% |
| | LogEIPC | 20.0% | 20.0% | 50.0% | 48.6% | 85.0% | 71.5% |

*Table 1.* Percentage of LCBench tasks ranked in the Top-$k$ ($k = 1, 2, 3$) by cost-adjusted simple regret for among acquisition–stopping rule pairs (excluding the hindsight stopping rule), for our PBGI/LogEIPC stopping rule paired with PBGI or LogEIPC, evaluated on large datasets (with $> 10,000$ instances) and on all datasets, for cost-scaling factor $\lambda \in \{10^{-3}, 10^{-4}, 10^{-5}\}$.

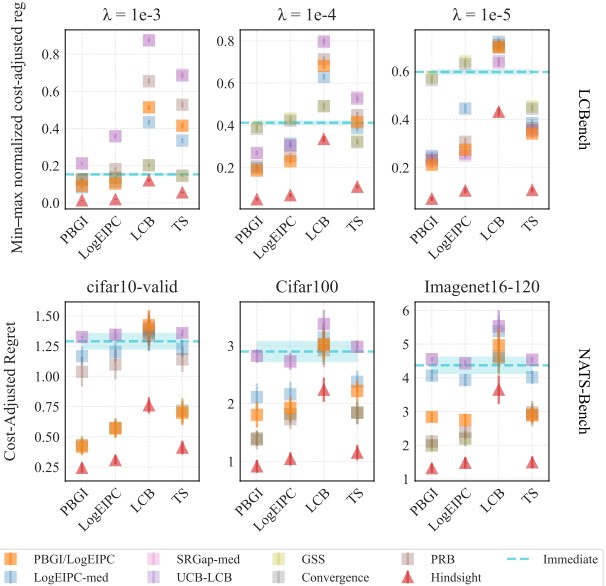

*Figure 3.* Cost-adjusted simple regret across acquisition function–stopping rule pairs on LCBench and NATS-Bench. The objective is to minimize validation error on classification tasks, with scaled proxy runtime as evaluation cost, scaled by representative values of $\lambda$ ($10^{-3}, 10^{-4}, 10^{-5}$ for LCBench and $10^{-5}$ for NATS-Bench). For LCBench, results are aggregated across 35 datasets using min–max normalization. Our PBGI/LogEIPC stopping rule, when paired with either PBGI or LogEIPC, typically ranks among the top 3 pairs and closely approaches the hindsight optimal on LCBench and on cifar10-valid in NATS-Bench, but slightly worse on the other two NATS datasets, likely due to model misspecification.

## 5. Conclusion

We develop the *PBGI/LogEIPC stopping rule* for Bayesian optimization with varying evaluation costs. Paired with either the PBGI or LogEIPC acquisition function, it (a) satisfies a theoretical guarantee bounding the expected cost-adjusted simple regret (Section 3.1), and (b) shows strong empirical performance in terms of cost-adjusted simple re-

gret (Section 4). We believe our framework can be extended to settings involving noisy, multi-fidelity or batched evaluations, as well as alternative objective formulations —for instance, applying a sigmoid transformation to test error rather than a linear one, to reflect real-world user preferences that shift sharply once error falls below a threshold.

## Impact Statement

This paper presents work whose goal is to advance the field of machine learning. There are many potential societal consequences of our work, none of which we feel must be specifically highlighted here.

## Acknowledgments

QX thanks Nairen Cao for helpful discussions on hindsight optimal, Theodore Brown for helpful discussions on LCBench runtime proxy, and Raul Astudillo for helpful discussions on unknown cost handling for LogEIPC. AT thanks James T. Wilson for helpful comments on stopping rules for Bayesian optimization. LC is funded by the European Union (ERC-2022-SYG-OCEAN-101071601). Views and opinions expressed are however those of the author(s) only and do not necessarily reflect those of the European Union or the European Research Council Executive Agency. Neither the European Union nor the granting authority can be held responsible for them. AT was supported by Cornell University, jointly via the Center for Data Science for Enterprise and Society, the College of Engineering, and the Ann S. Bowers College of Computing and Information Science. PF was supported by AFOSR FA9550-20-1-0351. ZS was supported by the NSF under grant no. CMMI-2307008.

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

# A. Theoretical Analysis and Calculations

In the following lemma, we prove a point-wise lower bound on the expected improvement before stopping for our recommended pairing of PBGI/LogEIPC stopping rule paired with either PBGI or LogEIPC acquisition function.[5]

**Lemma 3.1** (Expected improvement exceeds cost before stopping). *Let $X$ be compact, and let $f : X \to \mathbb{R}$ be a random function with prior mean $\mu(\cdot)$. Consider a Bayesian optimization algorithm that begins at some initial point $x_1 \in X$ with cost $C = c(x_1)$, acquires subsequent points using either the PBGI or LogEIPC acquisition function, and terminates according to the PBGI/LogEIPC stopping rule. Let $\tau = \min_{t \geq 1}\{\sup_{x \in X} \alpha_t^{\mathrm{LogEIPC}}(x) \leq 0\}$ be the algorithm's stopping time, and denote the posterior expected improvement function by $\alpha_t^{\mathrm{EI}}(x) = \mathrm{EI}_{f|x_{1:t}, y_{1:t}}(x; y_{1:t}^*)$. Then, for all $t < \tau$, $\alpha_t^{\mathrm{EI}}(x_{t+1}) \geq c(x_{t+1})$.*

*Proof.* While stopping has not occurred, meaning $t < \tau$, by the stopping criteria definition we have $\max_{x \in X} \alpha_t^{\mathrm{EI}}(x)/c(x) \geq 1$. Hence, there exists at least one point with $\alpha_t^{\mathrm{EI}}(x)/c(x) \geq 1$. We now argue for each algorithm.

*PBGI.* For each $x$ we defined a threshold $\alpha_t^{\mathrm{PBGI}}(x)$ by $\mathrm{EI}_{f|x_{1:t}, y_{1:t}}(x; \alpha_t^{\mathrm{PBGI}}(x)) = c(x)$. Since $\mathrm{EI}_{f|x_{1:t}, y_{1:t}}$ is increasing in its second argument,

$$y_{1:t}^* \geq \alpha_t^{\mathrm{PBGI}}(x) \iff \alpha_t^{\mathrm{EI}}(x)/c(x) \geq 1. \tag{13}$$

The existence of a point with ratio at least 1 therefore implies that the set $\mathcal{S}_t = \{x : y_{1:t}^* \geq \alpha_t^{\mathrm{PBGI}}(x)\}$ is non-empty. PBGI chooses $x_{t+1}$ with the *smallest* threshold, and thus

$$\alpha_t^{\mathrm{EI}}(x_{t+1}) = \mathrm{EI}_{f|x_{1:t}, y_{1:t}}(x_{t+1}; y_{1:t}^*) \geq \mathrm{EI}_{f|x_{1:t}, y_{1:t}}(x_{t+1}; \alpha_t^{\mathrm{PBGI}}(x_{t+1})) = c(x_{t+1}). \tag{14}$$

*(Log)EIPC.* By definition $x_{t+1}$ maximizes $\log(\alpha_t^{\mathrm{EI}}(x)/c(x))$ and $\alpha_t^{\mathrm{EI}}(x)/c(x)$, hence $\alpha_t^{\mathrm{EI}}(x_{t+1}) \geq c(x_{t+1})$.

Thus for *both* algorithms, we have

$$\alpha_t^{\mathrm{EI}}(x_{t+1}) \geq c(x_{t+1}), \qquad \text{for all } t < \tau. \tag{15}$$

$\square$

We can now use Lemma 3.1 to prove the following theorem, where we show that our PBGI/LogEIPC stopping rule (paired with the PBGI or LogEIPC acquisition function) also achieves cost-adjusted simple regret no worse than a naive baseline—stopping-immediately (*Immediate*).

**Theorem 3.2** (No worse than immediate stopping). *Consider the setting and algorithm specified in Lemma 3.1. Let $U := \mu(x_1) - \mathbb{E}[\min_{x \in X} f(x)] < \infty$, then the algorithm's expected cost-adjusted simple regret is bounded by*

$$\mathbb{E}\left[y_{1:\tau}^* - \min_{x \in X} f(x) + \sum_{t=1}^{\tau} c(x_t)\right]$$
$$\leq \mathbb{E}\big[\underbrace{y_1 - \min_{x \in X} f(x) + c(x_1)}_{\text{cost-adjusted regret of immediate stopping}}\big] = U + C. \tag{12}$$

*Proof.* We treat the two algorithms—meaning, the two acquisition function and stopping rule pairs—together and write $\mathcal{F}_t = \sigma(\{x_i, y_i\}_{i=1}^t)$ for the filtration generated by the observations. Since we are minimizing, the one–step improvement after iteration $t$ is $y_{1:t-1}^* - y_{1:t}^*$, where recall that $y_{1:t}^* = \min_{1 \leq i \leq t} y_i$. By our assumption about the random function $f$, $\mathbb{E}[y_{1:1}^*] = \mu(x_1)$ and the quantity $y_{1:1}^* - \min_{x \in X} f(x)$ has finite expectation $U < \infty$. Denote the posterior expected improvement function as $\alpha_t^{\mathrm{EI}}(x) = \mathrm{EI}_{f|x_{1:t}, y_{1:t}}(x; y_{1:t}^*)$.

By Lemma 3.1, $c(x_t) \leq \alpha_{t-1}^{\mathrm{EI}}(x_t)$ for all $t \leq \tau$. Set $\Delta_t = y_{1:t-1}^* - y_{1:t}^* \geq 0$ for $2 \leq t \leq \tau$. Conditional on $\mathcal{F}_{t-1}$ and the choice of $x_t$, we have

$$\mathbb{E}[\Delta_t \mid \mathcal{F}_{t-1}, x_t] = \alpha_{t-1}^{\mathrm{EI}}(x_t). \tag{16}$$

---

[5]In fact, any acquisition function (e.g., expected improvement-cost (EIC) (Hu et al., 2025)) that ensures the one-step expected improvement is worth the evaluation cost before the stopping time $\tau$ can apply here.

Taking expectations and summing up from 2 to $\tau$,

$$
\begin{aligned}
\mathbb{E}\left[\sum_{t=2}^{\tau} \alpha_{t-1}^{\mathrm{EI}}(x_t)\right] &= \mathbb{E}\left[\sum_{t=2}^{\infty} \mathbf{1}_{\{t \leq \tau\}} \, \alpha_{t-1}^{\mathrm{EI}}(x_t)\right] \\
&= \mathbb{E}\left[\sum_{t=2}^{\infty} \mathbf{1}_{\{t \leq \tau\}} \, \mathbb{E}[\Delta_t \mid \mathcal{F}_{t-1}, x_t]\right] \\
&= \mathbb{E}\left[\sum_{t=2}^{\infty} \mathbf{1}_{\{t \leq \tau\}} \, \Delta_t\right] \quad \text{(tower property)} \\
&= \mathbb{E}\left[\sum_{t=2}^{\tau} \Delta_t\right] = \mathbb{E}[y_{1:1}^* - y_{1:\tau}^*].
\end{aligned}
\tag{17}
$$

Summing (15) over $t \leq \tau$, taking expectations, and applying (17), we have

$$
\mathbb{E}\left[\sum_{t=2}^{\tau} c(x_t)\right] \leq \mathbb{E}\left[\sum_{t=2}^{\tau} \alpha_{t-1}^{\mathrm{EI}}(x_t)\right] \leq \mathbb{E}[y_{1:1}^* - y_{1:\tau}^*].
\tag{18}
$$

Adding the term $y_{1:\tau}^*$ on both sides of (18) gives

$$
\mathbb{E}\left[y_{1:\tau}^* + \sum_{t=2}^{\tau} c(x_t)\right] \leq \mathbb{E}[y_{1:1}^*].
$$

Finally, adding $c(x_1)$ to both sides and subtracting $\mathbb{E}[\min_{x \in X} f(x)]$ yields

$$
\begin{aligned}
\mathbb{E}\left[y_{1:\tau}^* - \min_{x \in X} f(x) + \sum_{t=1}^{\tau} c(x_t)\right] &\leq \mathbb{E}\left[y_{1:1}^* - \min_{x \in X} f(x) + c(x_1)\right] \\
&= \mathbb{E}\left[y_1 - \min_{x \in X} f(x) + c(x_1)\right] \\
&= \mu(x_1) - \mathbb{E}\left[\min_{x \in X} f(x)\right] + \mathbb{E}[c(x_1)] \\
&= U + C,
\end{aligned}
$$

since $\mathbb{E}[y_1] = \mu(x_1)$, $C = c(x_1)$, and $U = \mu(x_1) - \mathbb{E}[\min_{x \in X} f(x)]$. This is exactly (12). $\qquad\square$

*Remark* A.1. The quantities $U$ and $C$ depend on the choice of the initial point $x_1$. Choosing $x_1$ to minimize $\mu(x_1) + c(x_1)$ yields a tighter bound.

*Remark* A.2. In practice, one may center the objective by replacing $f(x)$ with $f(x) - \mu(x)$ and then using a zero-mean GP prior. This changes only the parameterization of the posterior mean (which can be shifted back by $\mu(x)$) and does not affect the implementation of the acquisition functions or stopping rules.

Notably, this guarantee may not hold for other acquisition–stopping rule pairings. Moreover, in the worst case, this is the best guarantee we can hope for—for instance, the evaluation costs can be uniformly high and no point is worth evaluating. We now formalize this worst-case regime.

**Proposition A.3** (Immediate stopping is optimal under large costs)**.** *Suppose the objective function is bounded as $f(x) \in [a, b]$ for all $x \in \mathcal{X}$, and define $B := b - a$. Consider any Bayesian optimization policy that starts from an initial evaluation $x_1$ and may take additional evaluations. If the evaluation cost satisfies*

$$
c(x) \geq B \quad \text{for all } x \in \mathcal{X},
$$

*then the policy that stops immediately after the initial evaluation is optimal with respect to expected cost-adjusted regret.*

*Proof.* Let $\pi_{\text{imm}}$ denote the policy that stops immediately after the initial evaluation $x_1$, and let $\pi$ be any other policy with stopping time $\tau \geq 1$. Recall that the cost-adjusted simple regret is

$$R_c(\pi) := \mathbb{E}\left[\min_{1 \leq t \leq \tau} f(x_t) - \min_{x \in \mathcal{X}} f(x) + \sum_{t=1}^{\tau} c(x_t)\right].$$

Since $y_1 = f(x_1)$ and $f(x) \in [a, b]$ for all $x \in \mathcal{X}$, we have

$$0 \leq y_1 - \min_{x \in \mathcal{X}} f(x) \leq b - a = B.$$

The immediate-stopping policy has $\tau = 1$, so

$$R_c(\pi_{\text{imm}}) = \mathbb{E}\left[y_1 - \min_{x \in \mathcal{X}} f(x) + c(x_1)\right].$$

Now consider an arbitrary policy $\pi$.

If $\tau = 1$, then $\pi$ coincides with immediate stopping and incurs the same cost-adjusted simple regret.

If $\tau \geq 2$, then the simple regret term is nonnegative, and the policy incurs at least one additional evaluation beyond $x_1$. Hence,

$$\min_{1 \leq t \leq \tau} f(x_t) - \min_{x \in \mathcal{X}} f(x) + \sum_{t=1}^{\tau} c(x_t) \geq c(x_1) + c(x_2).$$

Using $c(x) \geq B$ for all $x$,

$$c(x_1) + c(x_2) \geq c(x_1) + B \geq c(x_1) + \left(y_1 - \min_{x \in \mathcal{X}} f(x)\right).$$

Therefore, in all cases,

$$\min_{1 \leq t \leq \tau} f(x_t) - \min_{x \in \mathcal{X}} f(x) + \sum_{t=1}^{\tau} c(x_t) \geq y_1 - \min_{x \in \mathcal{X}} f(x) + c(x_1).$$

Taking expectations yields

$$R_c(\pi) \geq R_c(\pi_{\text{imm}}),$$

which proves optimality of immediate stopping. $\square$

**Corollary 3.3** (Finite-time termination under positive costs). *Consider the setting and algorithm specified in Lemma 3.1. Then the expected cumulative cost of the algorithm is bounded by $\mathbb{E}\left[\sum_{t=1}^{\tau} c(x_t)\right] \leq U + C$. Further, if evaluation costs are uniformly bounded below by a constant $c_0 > 0$, i.e., $c(x) \geq c_0, \forall x \in X$, then for any $\delta \in (0, 1)$, the algorithm terminates in at most $\frac{U+C}{\delta \cdot c_0}$ iterations with probability $1 - \delta$.*

*Proof.* Immediately by Theorem 3.2 and the fact that $y_{1:\tau}^* - \min_{x \in X} f(x) \geq 0$ pointwise, we have that the expected cumulative cost of the algorithm

$$\mathbb{E}\left[\sum_{t=1}^{\tau} c(x_t)\right] \leq U + C. \tag{19}$$

By Markov's inequality, for any $\delta \in (0, 1)$,

$$\Pr\left[\sum_{t=1}^{\tau} c(x_t) \leq \frac{U + C}{\delta}\right] \geq 1 - \delta.$$

Since $c(x_t) \geq c_0$ for all $t < \tau$, it follows that

$$\tau = \sum_{t=1}^{\tau} 1 \leq \sum_{t=1}^{\tau} \frac{c(x_t)}{c_0} = \frac{1}{c_0} \sum_{t=1}^{\tau} c(x_t).$$

Therefore,

$$\Pr\left[\tau \leq \frac{U+C}{\delta \cdot c_0}\right] \geq \Pr\left[\frac{1}{c_0}\sum_{t=1}^{\tau} c(x_t) \leq \frac{U+C}{\delta \cdot c_0}\right] = \Pr\left[\sum_{t=1}^{\tau} c(x_t) \leq \frac{U+C}{\delta}\right] \geq 1-\delta.$$

$\square$

**Corollary 3.5** (Choosing $\lambda$ for expected budget compliance)**.** *Consider the setting, algorithm, and notation specified in Lemma 3.1 and Theorem 3.2, but with costs rescaled by a factor $\lambda > 0$: both the acquisition values and the stopping conditions are computed using $\lambda c(\cdot)$. For some budget $B > C$, if the cost-scaling factor is set to $\lambda = \frac{U}{B-C}$, then the algorithm's expected cumulative unscaled cost satisfies $\mathbb{E}\left[\sum_{t=1}^{\tau} c(x_t)\right] \leq B$.*

*Proof.* Since the Bayesian optimization algorithm considers the post-scaling cost, by Theorem 3.2 and the fact that $y_{1:\tau}^* - \min_{x \in X} f(x) > 0$ pointwise, we have

$$\mathbb{E}\left[\sum_{t=1}^{\tau} \lambda \cdot c(x_t)\right] \leq \mathbb{E}\left[y_{1:\tau}^* - \min_{x \in X} f(x) + \sum_{t=1}^{\tau} \lambda \cdot c(x_t)\right] \leq \mu(x_1) - \mathbb{E}\left[\min_{x \in X} f(x)\right] + \lambda C = U + \lambda C. \quad (20)$$

Since $\lambda = U/(B-C)$, we have

$$\mathbb{E}\left[\sum_{t=1}^{\tau} c(x_t)\right] \leq \frac{U+\lambda C}{\lambda} = \frac{U}{\lambda} + C = B.$$

$\square$

*Remark* A.4. In the Bayesian regret setting where $f$ is drawn from a Gaussian process, i.e., $f \sim \mathcal{GP}(\mu(\cdot), K)$, the term $U = \mu(x_1) - \mathbb{E}\left[\min_{x \in X} f(x)\right]$ can be further bounded above using classical results on the expected supremum/infimum of Gaussian processes; see, for example, Lifshits (2012, Theorem 10.1). In many practical settings, the objective is naturally bounded. For example, in our AutoML experiments the objective is accuracy, which lies in $[0, 100]$. This implies

$$U = \mu(x_1) - \mathbb{E}\left[\min_{x \in X} f(x)\right] \leq 100,$$

providing a simple and interpretable upper bound on $U$.

## B. Experimental Setup

All experiments are implemented based on BoTorch (Balandat et al., 2020). Each Bayesian optimization procedure is initialized with $2(d+1)$ random samples, where $d$ is the dimension of the search domain. For Bayesian regret experiments, we follow the standard practice to generate the initial random samples using a quasirandom Sobol sequence. For empirical experiments, we randomly sample configuration IDs from a fixed pool of candidates—2,000 for LCBench and 32,768 for NATS-Bench. All computations are performed on CPU.

Each experiment is repeated with 50 random seeds, and we report the mean with error bars, given by two times the standard error, for each stopping rule. We also impose a cap on the number of iterations: 100 for 1D Bayesian regret, 500 for 8D Bayesian regret, and 200 for empirical experiments. If a stopping rule is not triggered before reaching this cap, we treat the stopping time as equal to the cap.

**Gaussian process models.** For all experiments, we follow the standard practice to apply Matérn kernels with smoothness $5/2$ and length scales learned from data via maximum marginal likelihood optimization, and standardize input variables to be in $[0, 1]^d$. For empirical experiments, we standardize output variables to be zero-mean and unit-variance, but not for Bayesian regret experiments. In this work, we consider the noiseless setting and set the fixed noise level to be $10^{-6}$. In the unknown-cost experiments, we follow Astudillo et al. (2021) to model the objective and the logarithm of the cost function using independent Gaussian processes.

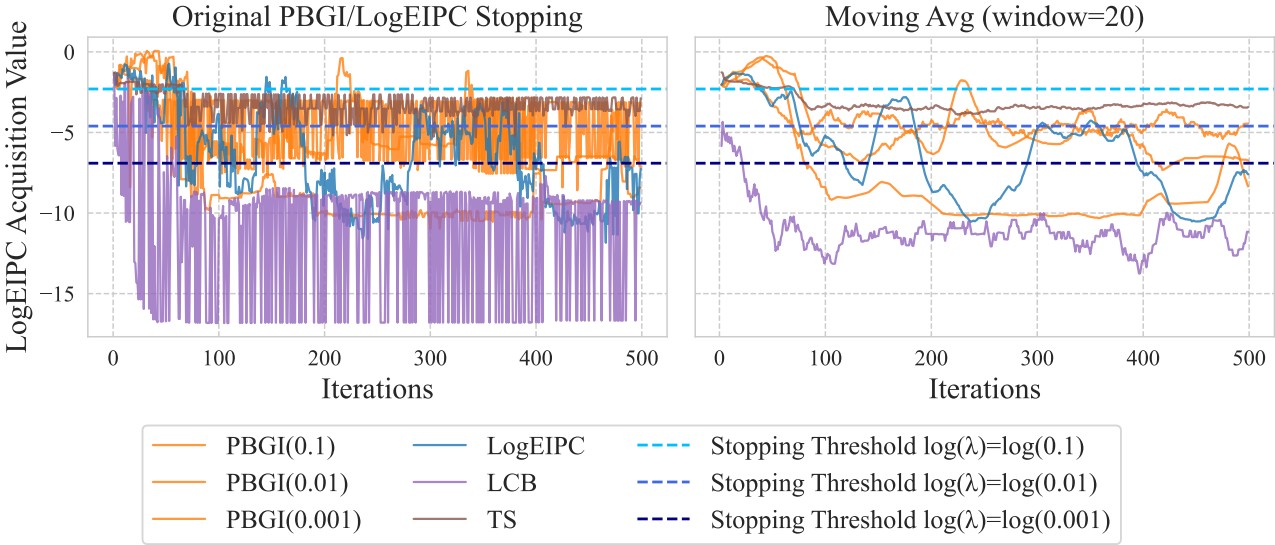

*Figure 4.* Comparison of the raw and moving-averaged PBGI/LogEIPC stopping rule signals (i.e., the LogEIPC acquisition values) in Bayesian regret 8D experiments for multiple acquisition functions, with linear cost and stopping thresholds at $\log(0.1)$, $\log(0.01)$ and $\log(0.001)$. *Left:* The unaveraged signals exhibits large, high-frequency fluctuations due to the difficulty of acquisition optimization in high dimensions. *Right:* Applying moving average (window=20) smooths these wiggles, yielding more stable stopping signals.

**Acquisition function optimization.** For the 1D Bayesian regret experiments, we optimize over 10,001 grid points. For the 8D Bayesian regret experiment, we use BoTorch's 'gen_candidates_torch', a gradient-based optimizer for continuous acquisition function maximization, as it avoids reproducibility issues caused by internal randomness in the default scipy optimizer. For the empirical experiments, since LCBench and NATS-Bench provide only 2,000 and 32,768 configurations respectively, we optimize the acquisition function by simply applying an argmin/argmax over the acquisition values of the unevaluated configurations, without using any gradient-based methods.

**Acquisition function and stopping rule parameters.** For PBGI, we follow Xie et al. (2024) to compute the Gittins indices using 100 iterations of bisection search without any early stopping or other performance and reliability optimizations.

For UCB/LCB-based acquisition functions and stopping rules, we follow the original GP-UCB paper Srinivas et al. (2009) and the choice in UCB/LCB based stopping rules (Makarova et al., 2022; Ishibashi et al., 2023) to use the schedule $\beta_t = 2\log(dt^2\pi^2/6\delta)$, where $d$ is the dimension. We also adopt their choice of $\delta = 10^{-1}$ and a scale-down factor of 5.

For PRB, we follow Wilson (2024) to use the schedule $N_t = \max(\lceil 64 * 1.5^{t-1}\rceil, 1000)$ for number of posterior samples, risk tolerance $\delta = 0.05$. The error bound $\epsilon$ is set to be 0.1 for Bayesian regret experiments and 0.5% of the best test error (here, the misclassification rate) among all configurations for the empirical experiments.

For the 8D Bayesian regret experiments, for all acquisition-value-based stopping rules, we apply moving average over 20 iterations to mitigate the fluctuations due to the imperfect acquisition function optimization. Figure 4 illustrates the challenges these oscillations pose when computing stopping rule statistics and shows the improvement with moving average. For consistency, we also apply the 20-iteration averaging to the non-acquisition-value-based GSS and Convergence baselines.

**Omitted baselines.** We omit the KG acquisition function and stopping rule due to its high computational cost, as they are shown to be computationally intensive in the runtime experiments of Xie et al. (2024).

**Objective functions: Bayesian regret.** In all Bayesian regret experiments, each objective function $f$ is sampled from a Gaussian process prior with a Matérn 5/2 kernel and a length scale of $0.1$, using a different seed in each of the 50 trials.

**Objective functions: empirical.** In the empirical experiments, the validation error (scaled out of 100) is used as the objective function during the Bayesian optimization procedure, while the cost-adjusted simple regret is reported based on the corresponding test error.

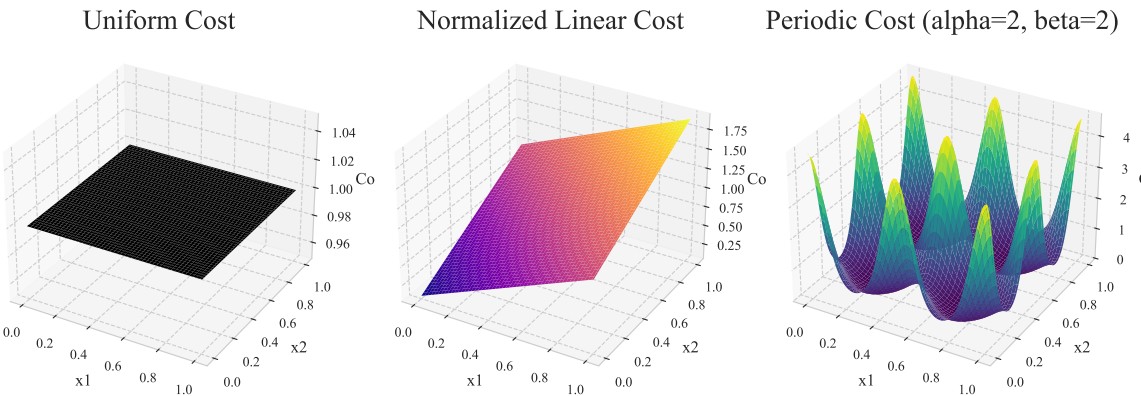

*Figure 5.* Surface plots of cost functions over $[0, 1]^2$: (Left) Uniform cost of 1 across domain. (Middle) Normalized linear cost increasing with the mean of $x_1$ and $x_2$. (Right) Periodic cost with $\alpha = 2$, $\beta = 2$, normalized by Bessel-based factor.

**Cost functions: Bayesian regret.** In Bayesian regret experiments, we consider three types of evaluation costs: uniform, linear, and periodic. These costs are normalized such that $\mathbb{E}_{x \in [0,1]^d}[c(x)]$ is approximately 1 and their expressions (prior to cost scaling) are given below.

In the *uniform* cost setting, each evaluation incurs a constant cost of 1.

In the *linear* cost setting, the cost increases proportionally with the average coordinate value of the input:

$$\text{linear\_cost}(x) = \frac{1 + 20 \cdot \left( \frac{1}{d} \sum_{i=1}^{d} x_i \right)}{11}.$$

In the *periodic* cost setting, the evaluation cost fluctuates across the domain. Following (Astudillo et al., 2021), we define the periodic cost as

$$\text{periodic\_cost}(x) = \frac{\exp \left( \frac{\alpha}{d} \sum_{i=1}^{d} \cos \left( 2\pi \beta (x_i - x_i^*) \right) \right)}{\left[ I_0 \left( \frac{\alpha}{d} \right) \right]^d},$$

where $x_i^*$ denotes the coordinate of the global optimum of $f$, and $I_0$ is the modified Bessel function of the first kind, which acts as a normalization constant. We set $\alpha = 2$ and $\beta = 2$ to induce noticeable variation in cost across the domain, while ensuring that costly evaluations can still be worthwhile.

A visualization of the three cost functions is provided in Figure 5.

**Cost functions: empirical.** In the *unknown-cost* experiments, we treat runtime—meaning, the provided full model training time (200 epochs for LCBench and 90 epochs for NATS)—as evaluation costs (prior to cost scaling).

In the *known-cost* experiments, for LCBench, we estimate the runtime cost from the number of model parameters $p$. Specifically, for the first three datasets in the six-dataset lite version, we use 0.001 times the number of model parameters as a proxy for runtime. This proxy is motivated by our observation of an approximately linear relationship between the number of model parameters and the actual runtime, with slope close to 0.001 (see Figure 6). For the full 35-dataset version, where the slope varies across datasets, we instead use a regression-derived coefficient $\alpha$ and the proxy cost is $\alpha p$. Importantly, the number of model parameters can be computed in advance, before the Bayesian optimization procedure, based on the network structure and classification task, as we explain in detail below.

In a feedforward neural network like shapedmlpnet with shape 'funnel', the model parameters are determined by input size (number of features), output size (e.g., number of classes), number of layers, size of each layer. The input size and output size are given by:

- Fashion-MNIST:
    - Input dimension: 784 (each image has 28×28 pixels, flattened into a vector of length 784)

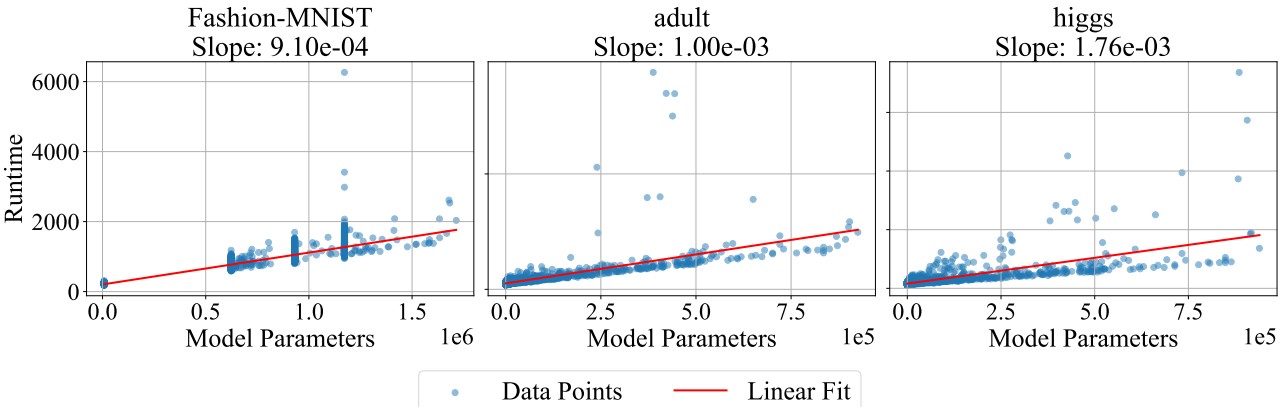

*Figure 6.* Empirical relationship between the number of model parameters and runtime for three LCBench datasets. Each subplot shows a scatter plot of actual runtime ($y$-axis) against number of model parameters ($x$-axis), along with a fitted linear regression line. The observed linear trend supports using 0.001 times the number of model parameters as a proxy for runtime. For *Fashion-MNIST* and *adult*, the fitted slopes are close to 0.001. The slope for higgs is slightly higher, possibly due to a few outliers.

      – Number of Output Features (output_feat): 10 (corresponding to 10 clothing categories)

- Adult:

      – Input Dimension: 14 (the dataset comprises 14 features, including both numerical and categorical attributes)
      – Number of Output Features: 1 (binary classification: income $> 50K$ or $\leq 50K$)

- Higgs:

      – Input Dimension: 28 (each instance has 28 numerical features)
      – Number of Output Features: 1 (binary classification: signal or background process)

The number of layers (num_layers) and size of each layer (max_unit) can be obtained from the configuration. With these information, we can compute the total number of model parameters (weights and biases) based on the layer-wise structure as follows:

$$\text{layer\_params}_{i \to i+1} = \text{layer}_i \cdot \text{layer}_{i+1} + \text{layer}_{i+1} \quad \text{(weights + biases)} \tag{21}$$

$$\text{model\_params} = \sum_{i=0}^{L-1} \text{layer\_params}_{i \to i+1} \tag{22}$$

$$\text{where} \quad \text{layer}_0 = \text{input\_dim}, \quad \text{layer}_L = \text{output\_feat} \tag{23}$$

Similarly for NATS-Bench, we use $\alpha \times F + \beta$ as a proxy for runtime (see Figure 7 for a visualization of the linear relationship), where $F$ is the number of floating point operations (FLOPs). Specifically, for *cifar10-valid*, we set $\alpha = 1$, $\beta = 400$; for *cifar100*, we set $\alpha = 2$, $\beta = 550$; and for *ImageNet16-120*, we set $\alpha = 1$, $\beta = 1000$.

FLOPs can also be computed in advance, as it is determined solely by the architecture's structure and the fixed input shape. Specifically, they are precomputed and stored for each architecture. Since each architecture corresponds to a deterministic computational graph and all inputs (e.g., CIFAR-10 images) have a fixed shape, the FLOPs required for a forward pass can be calculated analytically—without executing the model on data.

## C. Additional Experiment Results

In this section, we present additional experimental results to further evaluate the performance of different acquisition-function–stopping-rule pairs across various settings. We also include alternative visualizations to aid interpretation of the results.

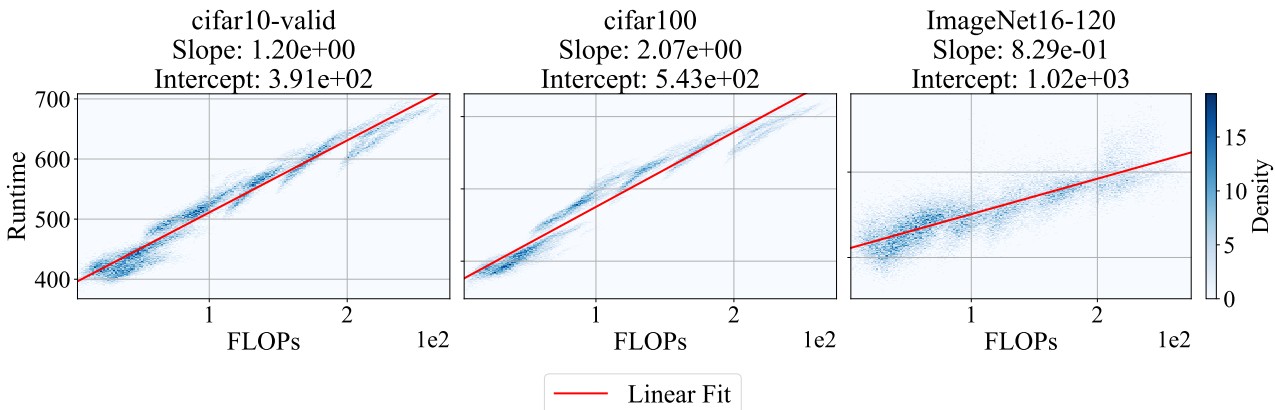

*Figure 7.* Empirical relationship between the number of FLOPs and runtime for the three NATS-Bench datasets. Each subplot shows a heatmap of actual runtime ($y$-axis) against number of FLOPs ($x$-axis), along with a fitted linear regression line. The observed linear trend supports using $\alpha \times \#\text{FLOPs} + \beta$ as a proxy for runtime.

### C.1. Runtime Comparison

First, we compare the runtime between our PBGI/LogEIPC stopping rule with several baselines. We measure the CPU time (in seconds) of the computation of the stopping rule, excluding the acquisition function computation and optimization.

From results in Figure 8 we can see that our PBGI/LogEIPC stopping rule is roughly as efficient as SRGap-med and UCB–LCB. In contrast, PRB is significantly more time-consuming, as it involves optimizing over up to 1000 samples.

### C.2. Order of Stopping and Posterior Updates

Following the discussions in Section 3, we always compute our proposed stopping rule with respect to the optimal acquisition function value of the *next round*—namely, the one which is obtained after posterior updates have been performed. One could alternatively consider checking the stopping criteria *before* posterior updates, which is backward-looking rather than forward-looking. Figure 9 provides an empirical comparison between the two choices, showing that stopping *after* the posterior update leads to stronger empirical performance.

This suggests that the theoretical guarantee for the Gittins index policy in the correlated Pandora's Box setting by Gergatsouli & Tzamos (2023), which is based on the *before-posterior-update* stopping, could potentially be improved by adopting the *after-posterior-update* stopping.

### C.3. Additional Experiment Results: Empirical

This subsection presents additional results for hyperparameter optimization on the LCBench datasets and neural archi­tecture–size search on the three NATS-Bench datasets. For LCBench, we provide results for the first three datasets from the six-dataset lite version of the benchmark: Fashion-MNIST, adult, and higgs under alternative settings (fixed budget, actual runtime, and unknown cost), along with an alternative visualization. We also include per-dataset results under the proxy-runtime setting for the full set of 35 datasets.

#### C.3.1. SIMPLE REGRET UNDER THE FIXED-BUDGET SETTING.

To isolate the effect of the acquisition function on cost-adjusted regret, we report the simple regret of several acquisition functions in the fixed-budget setting. We compare PBGI, LogEIPC, LCB, and TS, and additionally include PBGI-D with our recommended choice $\lambda_0 = U/(B - C)$ from Section 3.1.1, where $U = 50$ (reflecting the [0,100] range of classification accuracy) and $B - C = 10,000$ (corresponding to a budget after initial evaluation of 10,000 seconds, or roughly 3 hours). Cost-aware methods (the PBGI variants and LogEIPC) outperform cost-unaware ones (UCB and TS), with PBGI at smaller $\lambda$ consistently better than LogEIPC. This mirrors the findings of Xie et al. (2024) and helps explain the strong performance of the matched PBGI combination in our main experiments.

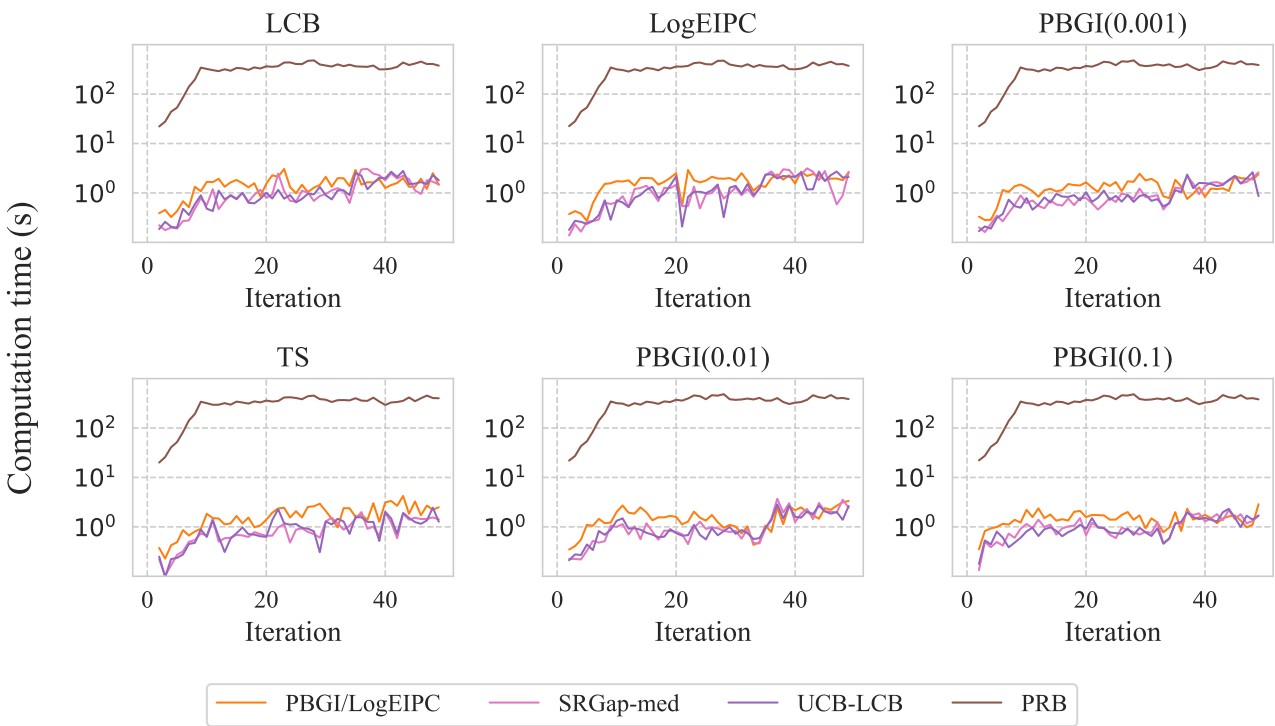

*Figure 8.* Evolution of per-iteration computation time (in log scale) for different stopping stopping rules when paired with six acquisition policies on the 8-dimensional Bayesian regret benchmark. Each subplot shows the average runtime (in seconds) over 50 iterations under one acquisition function—LCB, TS, LogEIPC, PBGI ($\lambda = 10^{-1}$), PBGI ($\lambda = 10^{-2}$), and PBGI ($\lambda = 10^{-3}$). Curves correspond to four stopping criteria: our PBGI/LogEIPC stopping rule, SRGap-med, UCB–LCB, and PRB. Convergence and GSS can be applied using only the best observed value and thus require no additional computation time, thus they are omitted here. LogEIPC-med relies on the same underlying statistical computations as the LogEIPC rule, and therefore its runtime is not measured separately. The results should that PRB incurs significant computational overhead compared to other stopping rules.

### C.3.2. NUMBER OF TRIALS WHERE STOPPING FAILS.

We count the number of trials in which a stopping rule fails to trigger within our iteration cap of 200 and present the results in Table 2. From the table, we observe that on datasets from the NATS benchmark, regret-based and acquisition-based stopping rules—except for ours—often fail to stop early. On LCBench datasets, some regret-based stopping rules such as SRGap-med and UCB–LCB also frequently exceed the cap. In contrast, our stopping rule consistently stops early, which aligns with our theoretical result in Corollary 3.5.

### C.3.3. ALTERNATIVE VISUALIZATION: COST-ADJUSTED SIMPLE REGRET VS ITERATION.

We provide an alternative visualization of cost-adjusted simple regret by plotting its mean and error bars at fixed iterations, along with the mean and error bars of the stopping iterations for each rule. This allows us to compare the performance of adaptive stopping rules not only against the hindsight-optimal adaptive stopping but also against the hindsight-optimal fixed-iteration stopping.

As shown in the empirical setting in Figures 11 to 16, cost-adjusted regret generally decreases in the early iterations and then increases. The turning point is exactly the hindsight-optimal fixed-iteration stopping point, and our PBGI/LogEIPC stopping rule consistently performs close to this optimum, particularly when paired with the PBGI acquisition function.

### C.3.4. COST MODEL MISSPECIFICATION: PROXY RUNTIME VS ACTUAL RUNTIME.

In the known-cost setting of hyperparameter tuning, a practical approach is to use a proxy for runtime as the evaluation cost during the Bayesian optimization procedure. In our case, we use the number of model parameters scaled by a constant factor, which can be known in advance and has been shown to correlate well with the actual runtime. However, for reporting

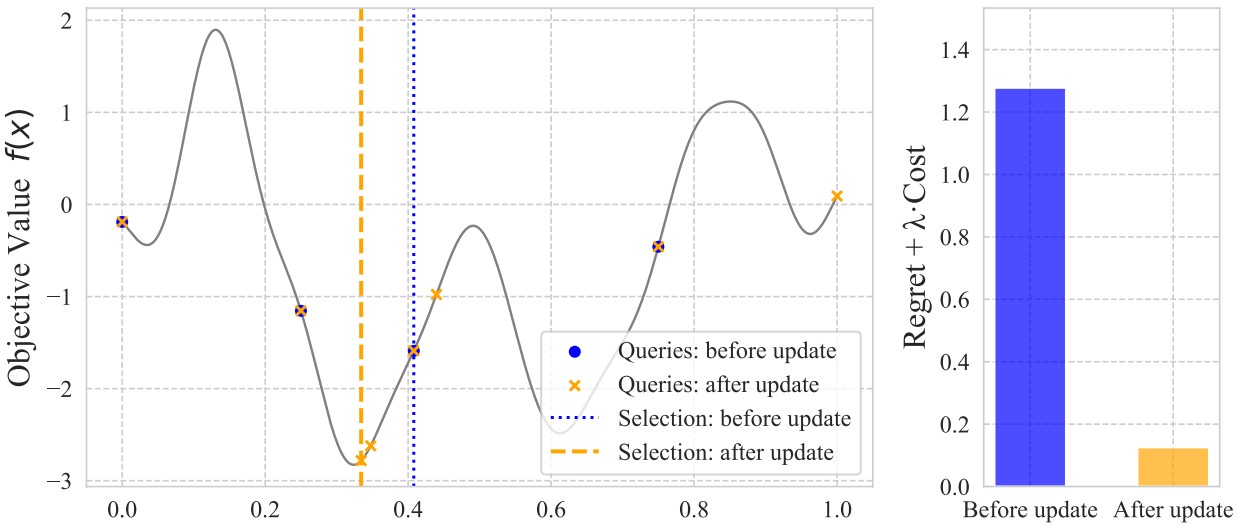

*Figure 9.* Illustration of a single draw from a Matérn-5/2 Gaussian process on $[0, 1]$ with lengthscale 0.1, optimized using PBGI acquisition function under uniform cost and cost-scaling factor $\lambda = 0.01$. We compare two variants of PBGI stopping rules: the *before-posterior-update* (this-round) stopping rule and the *after-posterior-update* (next-round) stopping rule. **Left:** The latent objective function (solid gray) and evaluation sequences for *before-posterior-update* stopping (blue circles) and *after-posterior-update* stopping (orange crosses). The dotted blue line and the dashed orange line mark the best observed value under each respective rule. **Right:** Cost-adjusted regret for each stopping rule. In this example, *after-posterior-update* stopping achieves strictly lower cost-adjusted regret despite performing more evaluations.

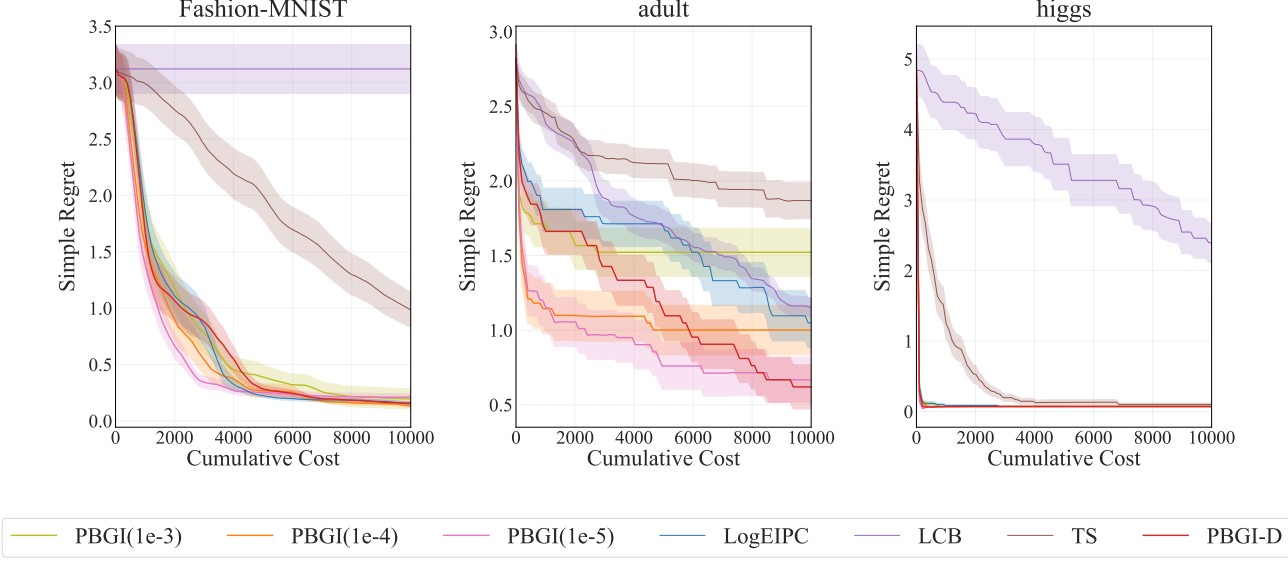

*Figure 10.* Comparison of simple regret of seven acquisition functions: PBGI ($\lambda = 10^{-3}$), PBGI ($\lambda = 10^{-4}$), PBGI ($\lambda = 10^{-4}$), LogEIPC, LCB, TS, and PBGI-D on LCBench datasets, using scaled proxy runtime as evaluation cost. We can see that indeed the cost-aware PBGI and LogEIPC outperform the cost-unaware UCB and TS. PBGI-D with our recommended $\lambda_0 = U/(B - C) = 50/10000 = 0.005$ is also competitive.

performance, one may prefer to use the actual runtime to better reflect real-world cost. To assess the impact of this cost model misspecification, we compare the cost-adjusted simple regret obtained when evaluation costs are computed using either the proxy runtime or the actual runtime. As shown in Figure 17, our PBGI/LogEIPC stopping rule remains close to the hindsight optimal even when there is a misspecification, although its ranking may shift slightly (e.g., from best to second-best on the *higgs* dataset).

*Table 2.* Number of trials (out of 50) where each stopping rule failed to trigger within 200 iterations, for each dataset in the LCBench (first three) and NATS (last three) benchmarks and each acquisition function. Results are identical across acquisition functions.

| Dataset | PBGI | LogEIPC-med | SRGap-med | UCB–LCB | GSS | Convergence | PRB |
|---|---|---|---|---|---|---|---|
| Fashion-MNIST | 0 | 0 | 0 | 50 | 0 | 0 | 0 |
| adult | 5 | 0 | 7 | 38 | 0 | 0 | 6 |
| higgs | 0 | 0 | 31 | 50 | 0 | 0 | 0 |
| Cifar10 | 0 | 32 | 50 | 50 | 0 | 0 | 26 |
| Cifar100 | 3 | 17 | 50 | 50 | 0 | 0 | 2 |
| ImageNet | 0 | 23 | 50 | 50 | 0 | 0 | 0 |

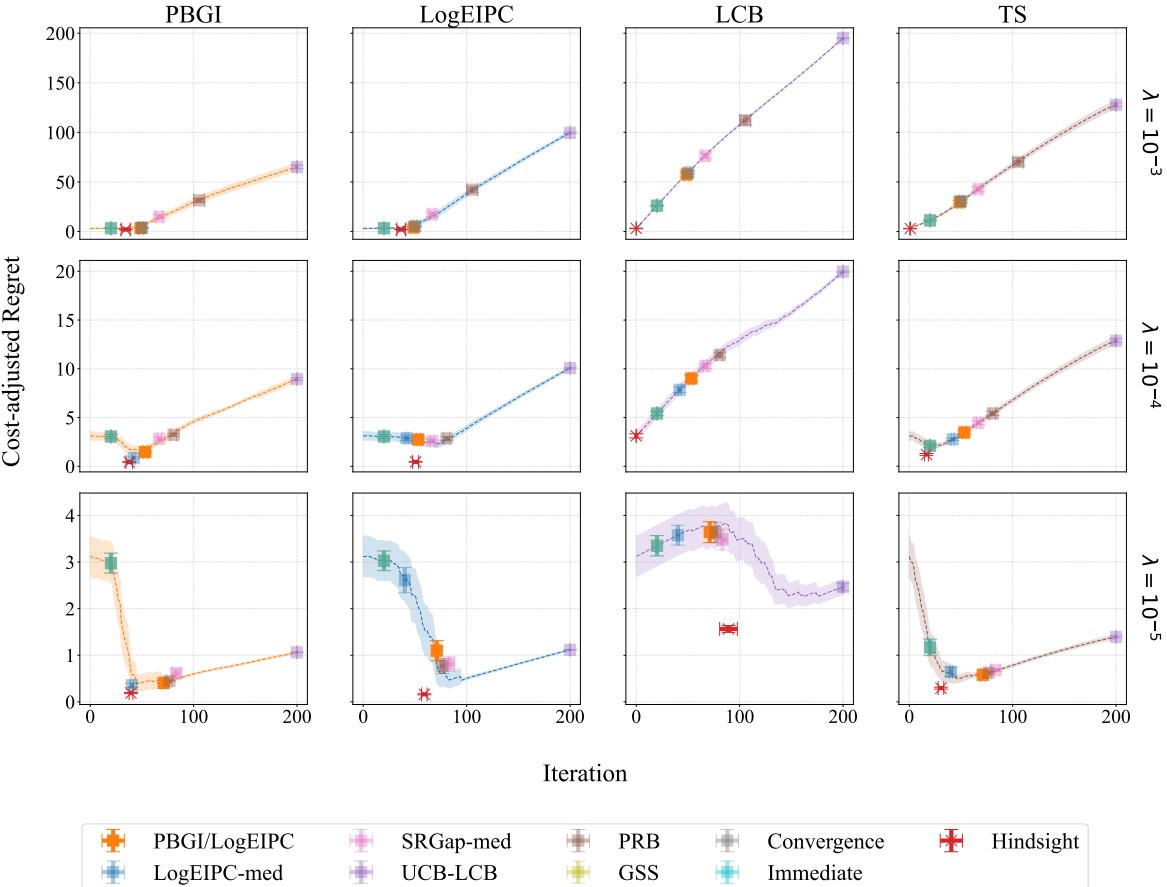

*Figure 11.* Comparison of cost-adjusted simple regret across acquisition function and stopping rule pairs on the *Fashion-MNIST* dataset. The objective function is the validation error, and the evaluation cost is the proxy runtime, scaled by three different cost-scaling factors $\lambda = 10^{-3}, 10^{-4}, 10^{-5}$. We can see that the PBGI/LogEIPC stopping rule consistently achieves cost-adjusted regret close to the hindsight optimal adaptive stopping (shown by the red marker) as well as the hindsight optimal fixed-iteration stopping (given by the minimum point along the fixed-budget curve) when paired with the PBGI acquisition function, though not always the best.

**Unknown-cost.** Astudillo et al. (2021, Proposition 2) proposed modeling unknown cost $c(x)$ via

$$\mathbb{E}[1/c(x)]^{-1} = \exp(\mu_{\ln c}(x) - (\sigma_{\ln c}(x))^2/2). \tag{24}$$

An alternative is

$$\mathbb{E}[c(x)] = \exp(\mu_{\ln c}(x) + (\sigma_{\ln c}(x))^2/2). \tag{25}$$

The difference in sign before the variance term reflects how each formulation handles predictive uncertainty: (24) encourages more exploration than (25). For PBGI under the unknown-cost setting, it is more natural to replace $c(x)$ in (4) with $\mathbb{E}[c(x)]$

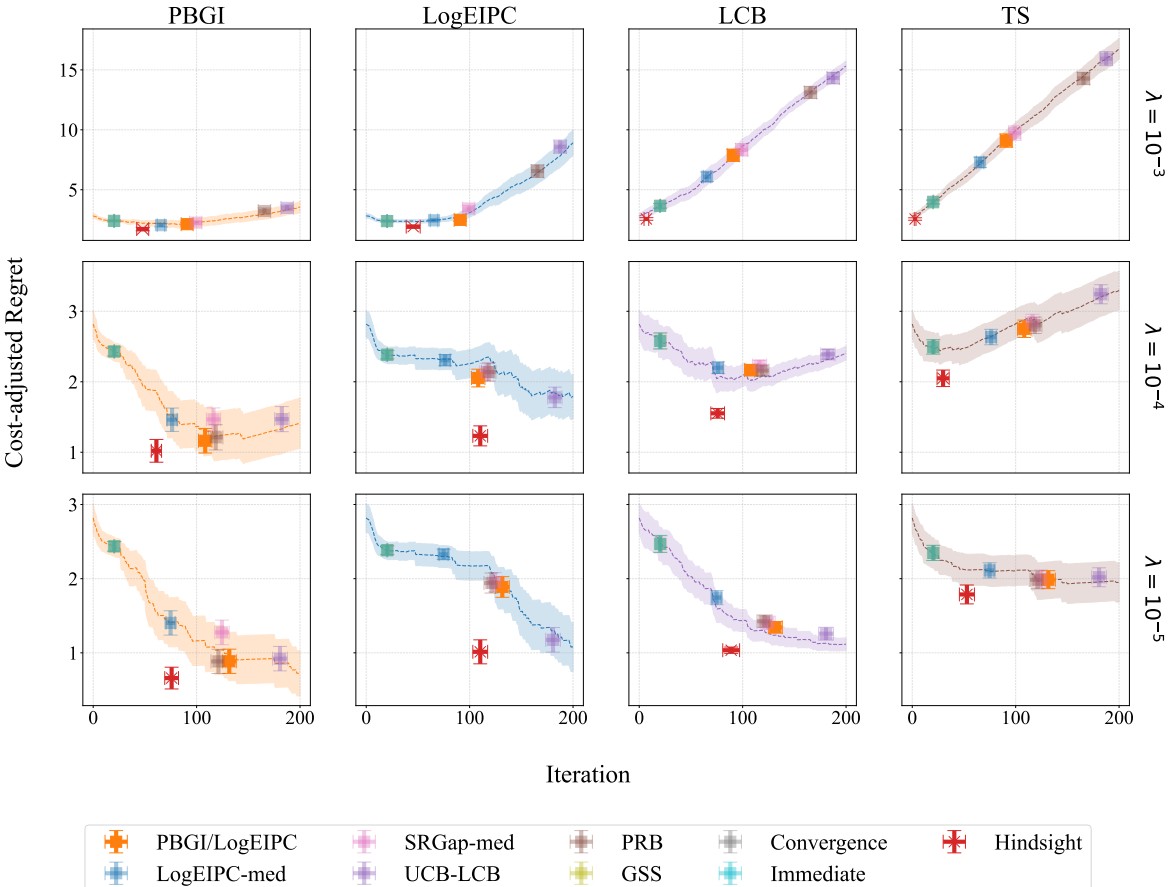

*Figure 12.* Comparison of cost-adjusted simple regret across acquisition function and stopping rule pairs on the *adult* dataset. The objective function is the validation error, and the evaluation cost is the proxy runtime, scaled by three different cost-scaling factors $\lambda = 10^{-3}, 10^{-4}, 10^{-5}$. We can see that the PBGI/LogEIPC stopping rule consistently achieves the cost-adjusted regret close to hindsight optimal adaptive stopping (shown by the red marker) as well as the hindsight optimal fixed-iteration stopping (given by the minimum point along the fixed-budget curve) when paired with the PBGI or TS acquisition function, particularly with PBGI.

using (25), as this aligns with how costs enter the root-finding problem. For LogEIPC, both variants are possible—we refer to the (24) version as *LogEIPC-inv* and the (25) version as *LogEIPC-exp*. However, equivalence between PBGI and LogEIPC stopping rules and our theoretical guarantees hold only with (25) but not with (24). Accordingly, we use (25) for both methods in our experiments to maintain consistency and preserve this equivalence. Figure 18 shows performance of acquisition function and stopping rule pairs under the unknown-cost setting, which are qualitatively similar to the known-cost setting.

### C.3.5. COST-ADJUSTED SIMPLE REGRET OF ALL LCBENCH DATASETS.

Due to space constraints, for LCBench, Figure 3 in the main text reports min–max normalized cost-adjusted simple regret, defined as

$$\frac{r - r_{\min}}{r_{\max} - r_{\min}},$$

where $r$ denotes cost-adjusted regret of a given acquisition function–stopping rule pair, and $r_{\min}$ and $r_{\max}$ are the minimum (including the hindsight optimal) and maximum cost-adjusted regret across all pairs. This normalization enables aggregation across datasets with different regret scales.

Here we present the *unnormalized* bar-plot results for each LCBench dataset (OpenML dataset size in parentheses) under all three cost-scaling parameters in Figures 19 to 21. As discussed in the main text, our PBGI/LogEIPC stopping rule—when

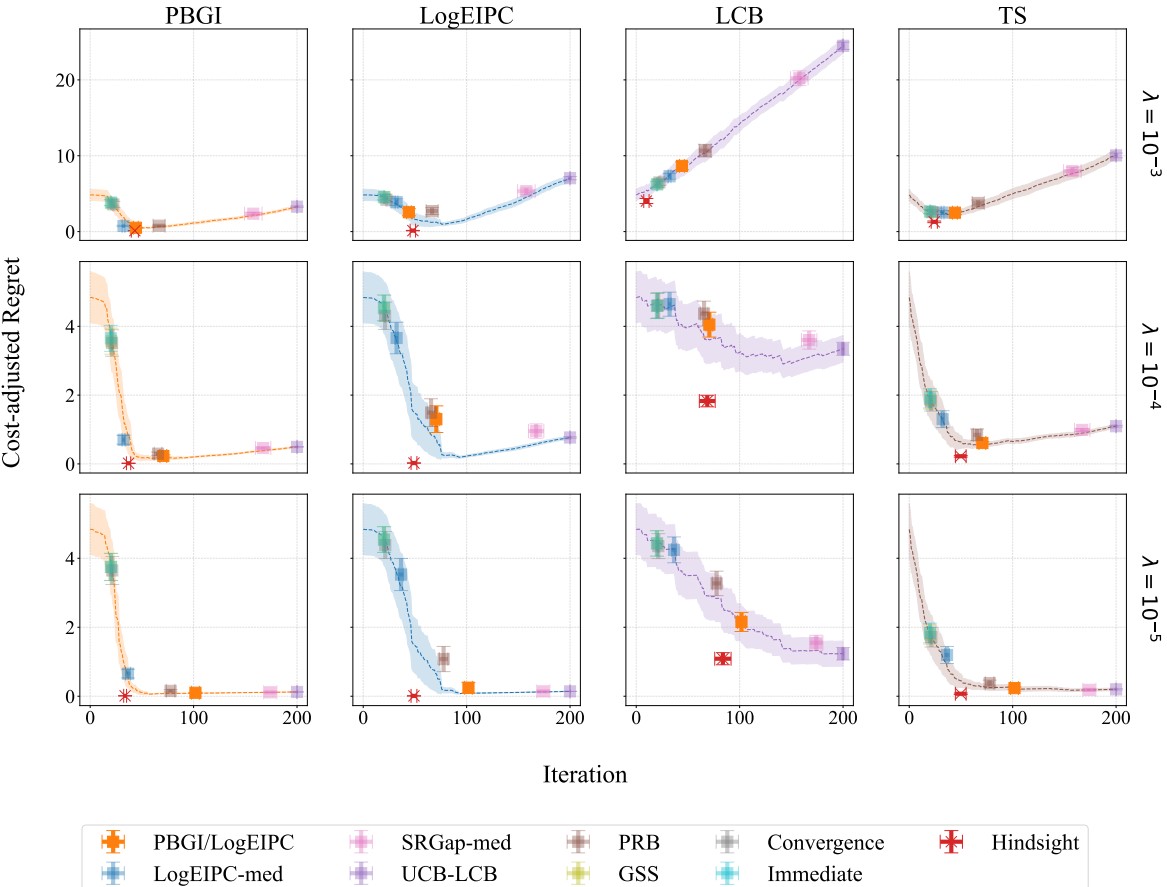

*Figure 13.* Comparison of cost-adjusted simple regret across acquisition function and stopping rule pairs on the *higgs* dataset. The objective function is the validation error, and the evaluation cost is the proxy runtime, scaled by three different cost-scaling factors $\lambda = 10^{-3}, 10^{-4}, 10^{-5}$. We can see that the PBGI/LogEIPC stopping rule consistently achieves the best cost-adjusted regret when paired with the PBGI, LogEIPC, or TS acquisition function, particularly with PBGI. These pairs not only approach the hindsight optimal adaptive stopping (shown by the red marker) but also perform comparably to the hindsight optimal fixed-iteration stopping (the minimum point along the fixed-budget curve).

paired with either PBGI or LogEIPC—achieves competitive performance on roughly 75% of the 35 datasets. However, the matched PBGI pair consistently underperforms across all three $\lambda$ values on the following datasets (see subplots with light grey background in Figures 19 to 21): Amazon_employee_access (32769), Australian (690), KDDCup09_appetency (50000), cnae-9 (1080), credit-g (1000), fabert (8237), and vehicle (846). Three additional datasets, jasmine (2984), mfeat-factors (2000) and sylvine (5124), show acceptable performance only when $\lambda = 10^{-5}$.

With the exception of Amazon_employee_access (32769) and KDDCup09_appetency (50000), all datasets on which the matched PBGI pair underperforms have fewer than 10,000 instances. This suggests that relatively small dataset size may contribute to model misspecification and degraded performance.

### C.3.6. DIAGNOSING UNDERPERFORMING LCBENCH DATASETS VIA VALIDATION–TEST RANK PRESERVATION.

We show that the degraded performance of our stopping rule on smaller LCBench datasets is likely caused by a benchmark-intrinsic val/test mismatch rather than a deficiency of the rule itself. We classify a dataset as "underperforming" if the cost-adjusted simple regret (at $\lambda = 10^{-4}$) of the matched PBGI pair does not rank among the top three PBGI-based acquisition function–stopping rule pairs formed by the seven non-trivial stopping rules we study. Formally, let $\mathcal{S} = \{\text{PBGI/LogEIPC}, \text{LogEIPC-med}, \text{SRGap-med}, \text{UCB–LCB}, \text{PRB}, \text{GSS}, \text{Convergence}\}$ (excluding the Immediate and Hindsight baselines), and let $r_d(s) := \mathcal{R}_c^{(\lambda=10^{-4})}(\text{PBGI}, s; d)$ denote the mean cost-adjusted regret of PBGI paired with

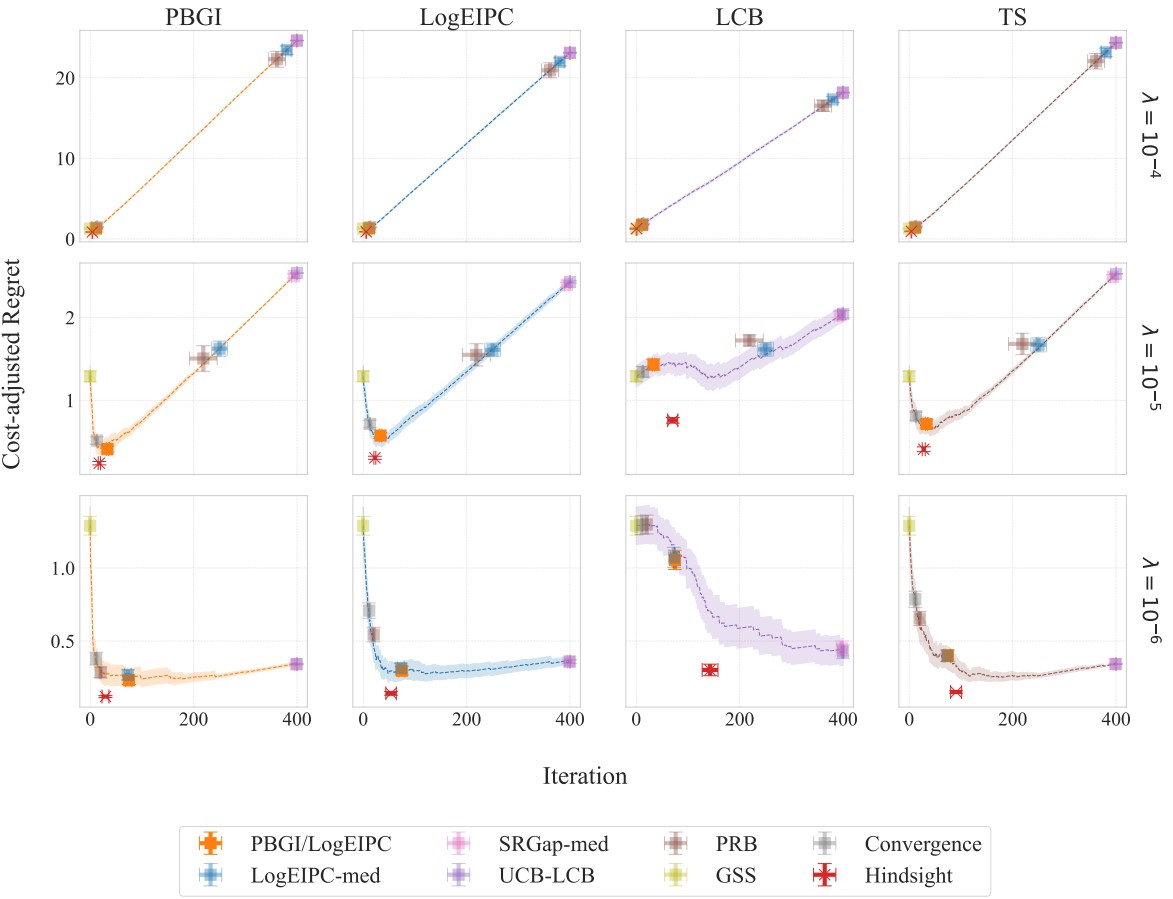

*Figure 14.* Comparison of cost-adjusted simple regret across acquisition function and stopping rule pairs on the *cifar10-valid* dataset. The objective function is the validation error, and the evaluation cost is the proxy runtime, scaled by three different cost-scaling factors $\lambda = 10^{-4}, 10^{-5}, 10^{-6}$. We can see that the PBGI/LogEIPC stopping rule consistently achieves the best cost-adjusted regret when paired with the PBGI, LogEIPC, or TS acquisition function, particularly with PBGI. These pairs not only approach the hindsight optimal adaptive stopping but also perform comparably to the hindsight optimal fixed-iteration stopping.

rule $s$ on dataset $d$. With $\rho_d := \left| \{ s \in \mathcal{S} : r_d(s) \leq r_d(\text{PBGI/LogEIPC}) \} \right|$ the within-dataset rank of the matched rule (rank 1 best), we set

$$d \in \begin{cases} \mathcal{D}_{\text{top}} & \text{if } \rho_d \leq 3, \\ \mathcal{D}_{\text{under}} & \text{if } \rho_d \geq 4. \end{cases} \quad (26)$$

The LCBench experiments optimize validation error but report test error. We quantify how faithfully validation ranks transfer to test ranks via the top-$k$ overlap fraction. For dataset $d$ with $n_d$ configurations, let $\pi_d^{\text{val}}, \pi_d^{\text{test}} \in S_{n_d}$ be the rankings of configurations by validation and test error (ascending, best first); then

$$\text{Top-}k(d) := \frac{1}{k} \left| \{ \pi_d^{\text{val}}(1), \ldots, \pi_d^{\text{val}}(k) \} \cap \{ \pi_d^{\text{test}}(1), \ldots, \pi_d^{\text{test}}(k) \} \right|. \quad (27)$$

We take $k = 20$, comparable to the 200-iteration cap in our experiments and to the top tail of configurations any adaptive method realistically considers as stopping candidates.

Figure 22 reports Top-20$(d)$ for every LCBench dataset, color-coded by the above classification. The $\mathcal{D}_{\text{under}}$ datasets have *substantially* lower val–test rank preservation than those in $\mathcal{D}_{\text{top}}$:

$$\overline{\text{Top-20}}\Big|_{\mathcal{D}_{\text{under}}} = \mathbf{36.1}\% \ (n = 14, \ \text{std} = 35.7\%) \quad \ll \quad \overline{\text{Top-20}}\Big|_{\mathcal{D}_{\text{top}}} = \mathbf{83.6}\% \ (n = 21, \ \text{std} = 23.8\%).$$

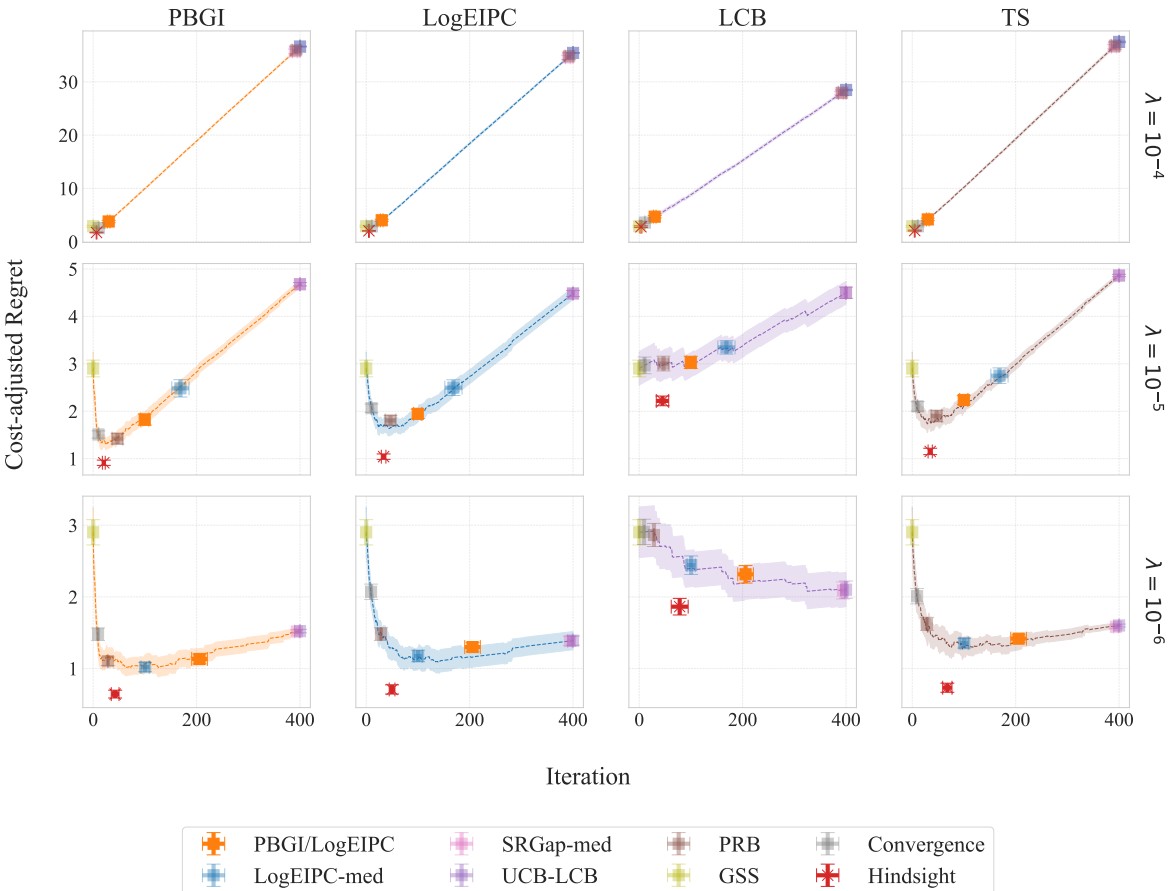

*Figure 15.* Comparison of cost-adjusted simple regret across acquisition function and stopping rule pairs on the *cifar100* dataset. The objective function is the validation error, and the evaluation cost is the proxy runtime, scaled by three different cost-scaling factors $\lambda = 10^{-4}, 10^{-5}, 10^{-6}$. The PBGI/LogEIPC stopping rule remains competitive at $\lambda = 10^{-4}$ and $10^{-6}$, though not always the best. At $\lambda = 10^{-5}$, unlike in most experiments, it stops noticeably late (though still outperforming several other rules), even when paired with its matching acquisition function. By Lemma 3.1, under model match, the PBGI/LogEIPC rule with the corresponding acquisition function should never incur negative expected cost-adjusted regret before stopping. The suboptimal behavior observed here points to significant model misspecification on the *cifar100* dataset.

When the validation-selected top-$k$ shares little overlap with the test top-$k$, no validation-driven stopping rule can recover test-optimal configurations, so the matched PBGI pairing's apparent underperformance is best attributed to the benchmark's intrinsic val/test mismatch rather than to the stopping rule.

### C.4. Additional Experiment Results: Bayesian Regret

In this section, we present the complete Bayesian regret results. Figures 23 to 25 show the 1D experiments, and Figures 26 to 28 show the 8D experiments. Each figure corresponds to one cost setting (uniform, linear or periodic) and three values of the cost-scaling factor, $\lambda = 10^{-1}, 10^{-2}, 10^{-3}$. In all of the experimental results, we observe that PBGI/LogEIPC acquisition function + PBGI/LogEIPC stopping achieves cost-adjusted regret that is not only competitive with the baselines, but is also competitive regarding the *best in hindsight* fixed iteration stopping and often competitive even comparing to *hindsight optimal* stopping. These results indicate that our automatic stopping rule can replace manual selection of stopping times without loss in performance. Figures 23 to 28 also show that our PBGI/LogEIPC stopping rule outperforms other baselines when the cost-scaling factor $\lambda$ is large, indicating that it's an especially suitable stopping criteria when evaluation is expensive.

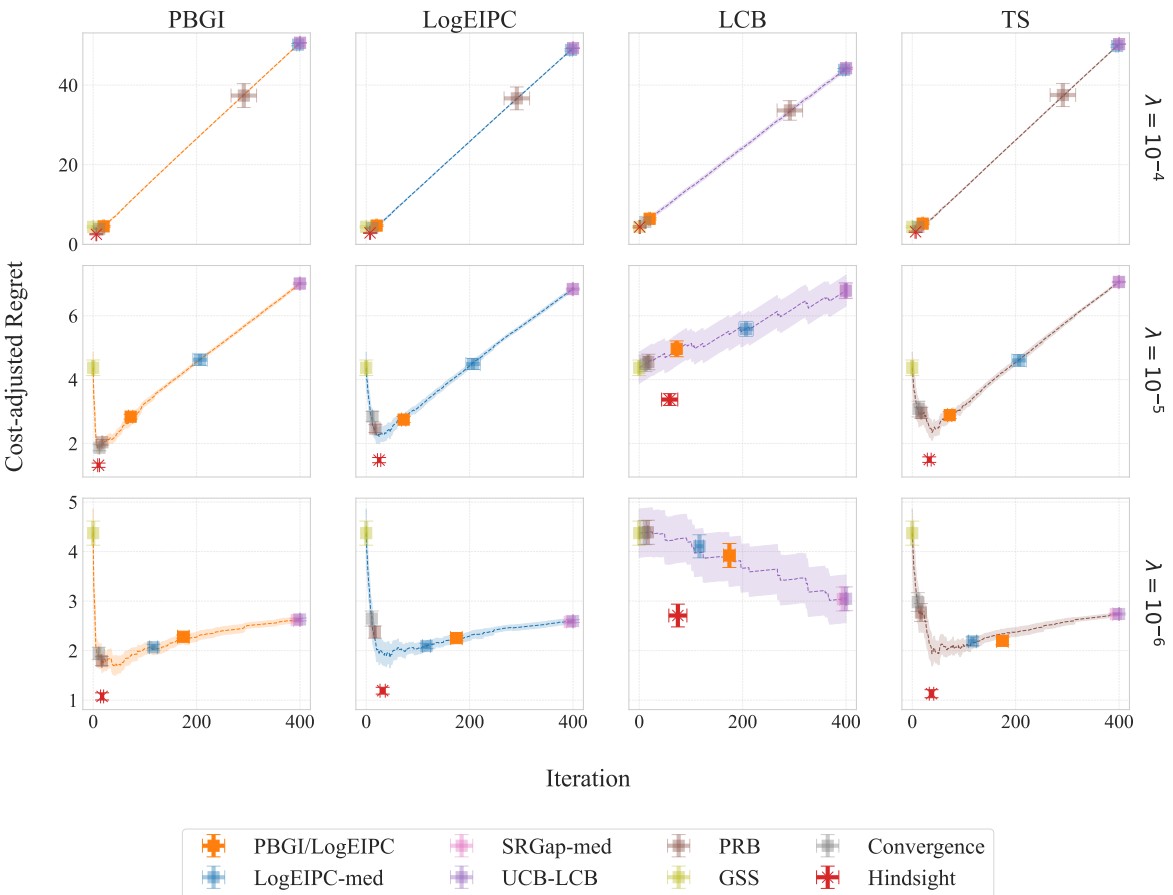

*Figure 16.* Comparison of cost-adjusted simple regret across acquisition function and stopping rule pairs on the *ImageNet16-120* dataset. The objective function is the validation error, and the evaluation cost is the proxy runtime, scaled by three different cost-scaling factors $\lambda = 10^{-4}, 10^{-5}, 10^{-6}$. The PBGI/LogEIPC stopping rule remains competitive at $\lambda = 10^{-4}$ and $10^{-6}$, though not always the best. At $\lambda = 10^{-5}$, unlike in most experiments, it stops noticeably late (though still outperforming several other rules), even when paired with its matching acquisition function. By Lemma 3.1, under model match, the PBGI/LogEIPC rule with the corresponding acquisition function should never incur negative expected cost-adjusted regret before stopping. The suboptimal behavior observed here points to significant model misspecification on the *ImageNet16-120* dataset.

### C.4.1. ABLATION: MOVING-AVERAGE WINDOW SIZE

We ablate the window size $W \in \{1, 5, 10, 20\}$ for the PBGI/LogEIPC stopping rule on the 8D Bayesian regret benchmark, where $W = 1$ corresponds to no smoothing. Figures 29 to 32 show how the window size affects the stopping iteration and the resulting cost-adjusted regret along a single acquisition-function path under each cost regime. Aggregate results across 50 seeds for the PBGI/LogEIPC stopping rule paired with its matched acquisition function under the three cost regimes (uniform, linear, periodic) are summarized in Figures 33 to 35.

Across all three cost regimes, applying a moving average consistently improves cost-adjusted regret when the PBGI/LogEIPC stopping rule is paired with its matched acquisition function, with the largest gains observed under periodic costs.

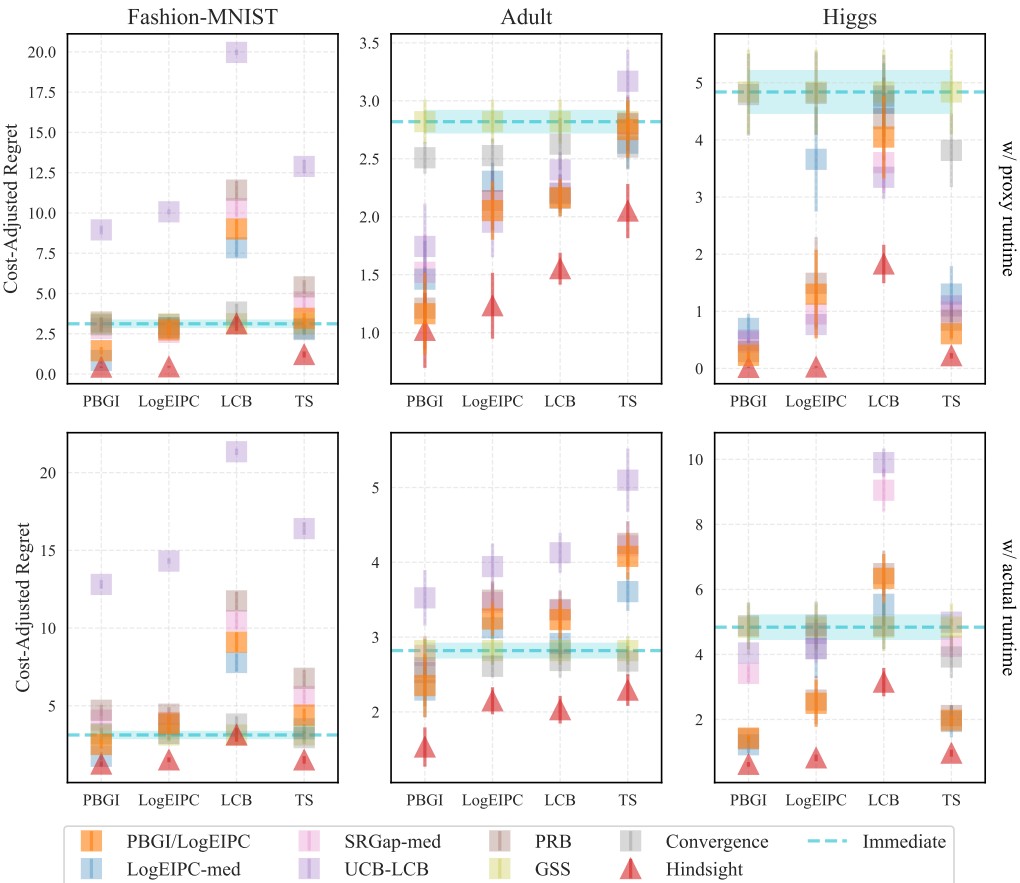

*Figure 17.* Comparison of cost-adjusted simple regret on three LCBench datasets with $\lambda = 10^{-4}$, using scaled proxy runtime vs. scaled actual runtime as evaluation cost. While our PBGI/LogEIPC stopping rule performs slightly worse under actual runtime (e.g., dropping from best to second-best on the *higgs* dataset), likely due to cost model misspecification, it remains close to the hindsight optimal in all cases.

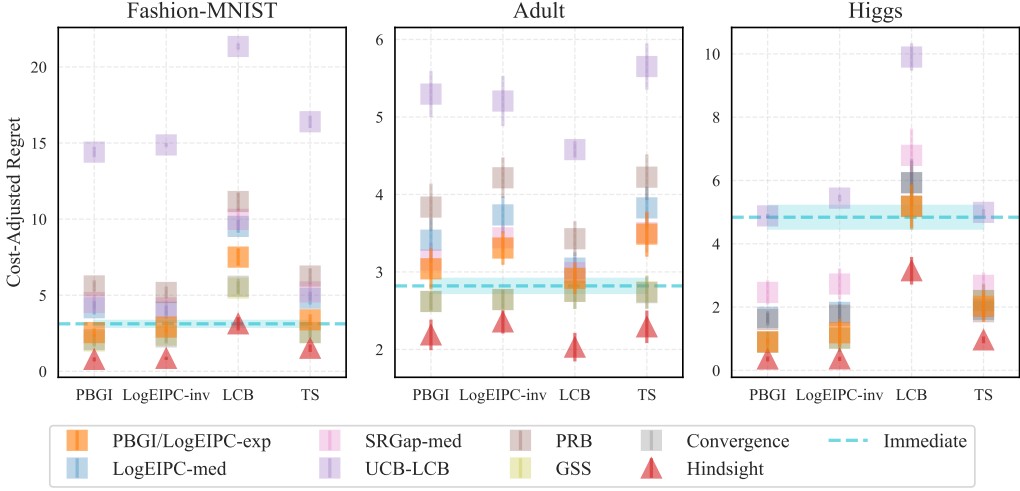

*Figure 18.* Cost-adjusted simple regret across acquisition function and stopping rule pairs under the unknown-cost setting on LCBench with $\lambda = 10^{-4}$. Our PBGI/LogEIPC stopping rule remains close to the hindsight optimal when paired with the PBGI or LogEIPC-exp acquisition function, sometimes slightly worse than heuristics such as GSS and Convergence. It is also slightly worse than Immediate on *Adult*, likely due to a cost-model misspecification in the unknown-cost setting.

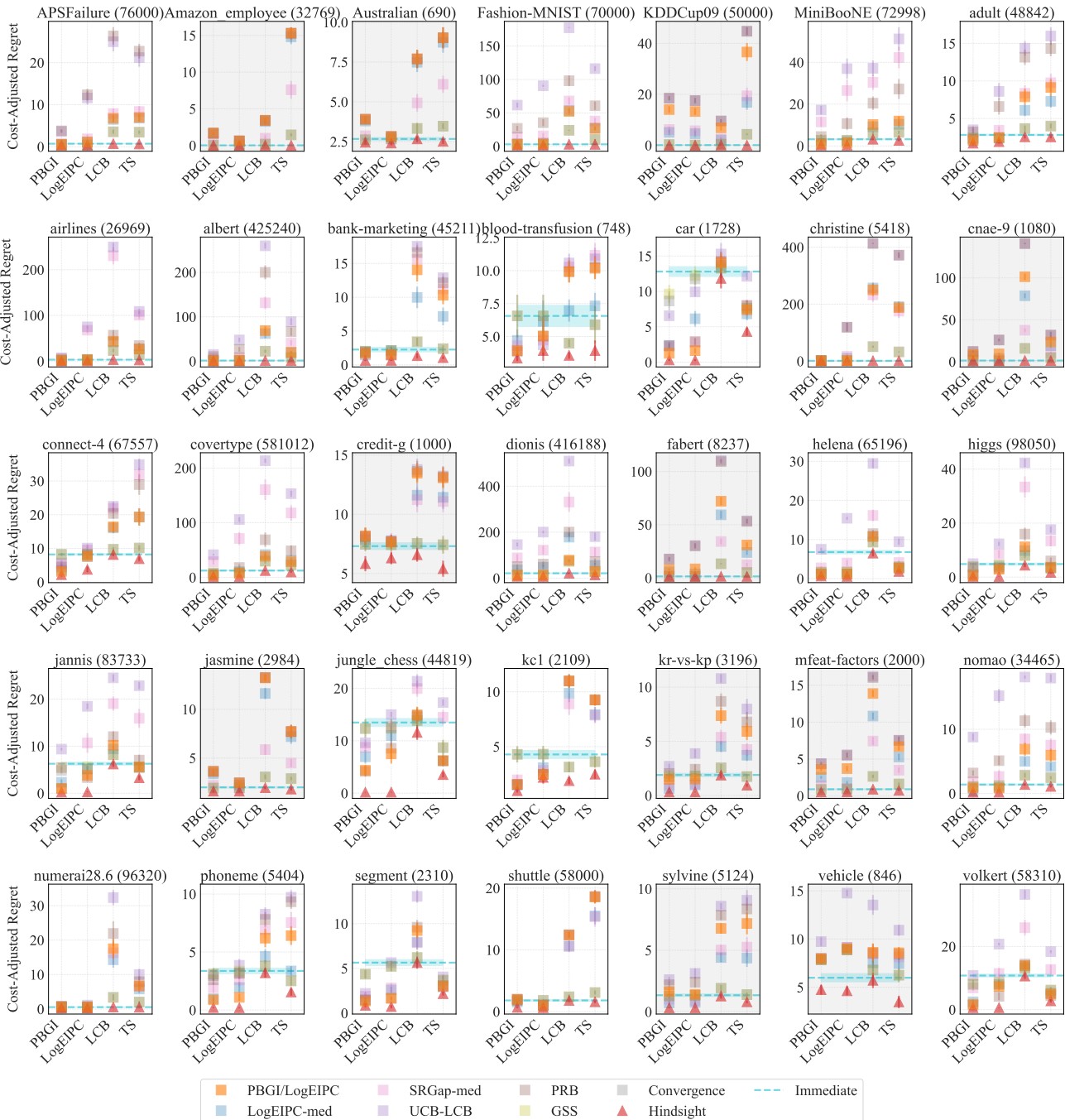

*Figure 19.* Cost-adjusted simple regret of all 35 LCBench datasets when $\lambda = 10^{-3}$. Datasets where the matched PBGI pair underperforms, meaning that it does not rank among the top three acquisition function–stopping rule pairs by cost-adjusted simple regret, are highlighted in light grey.

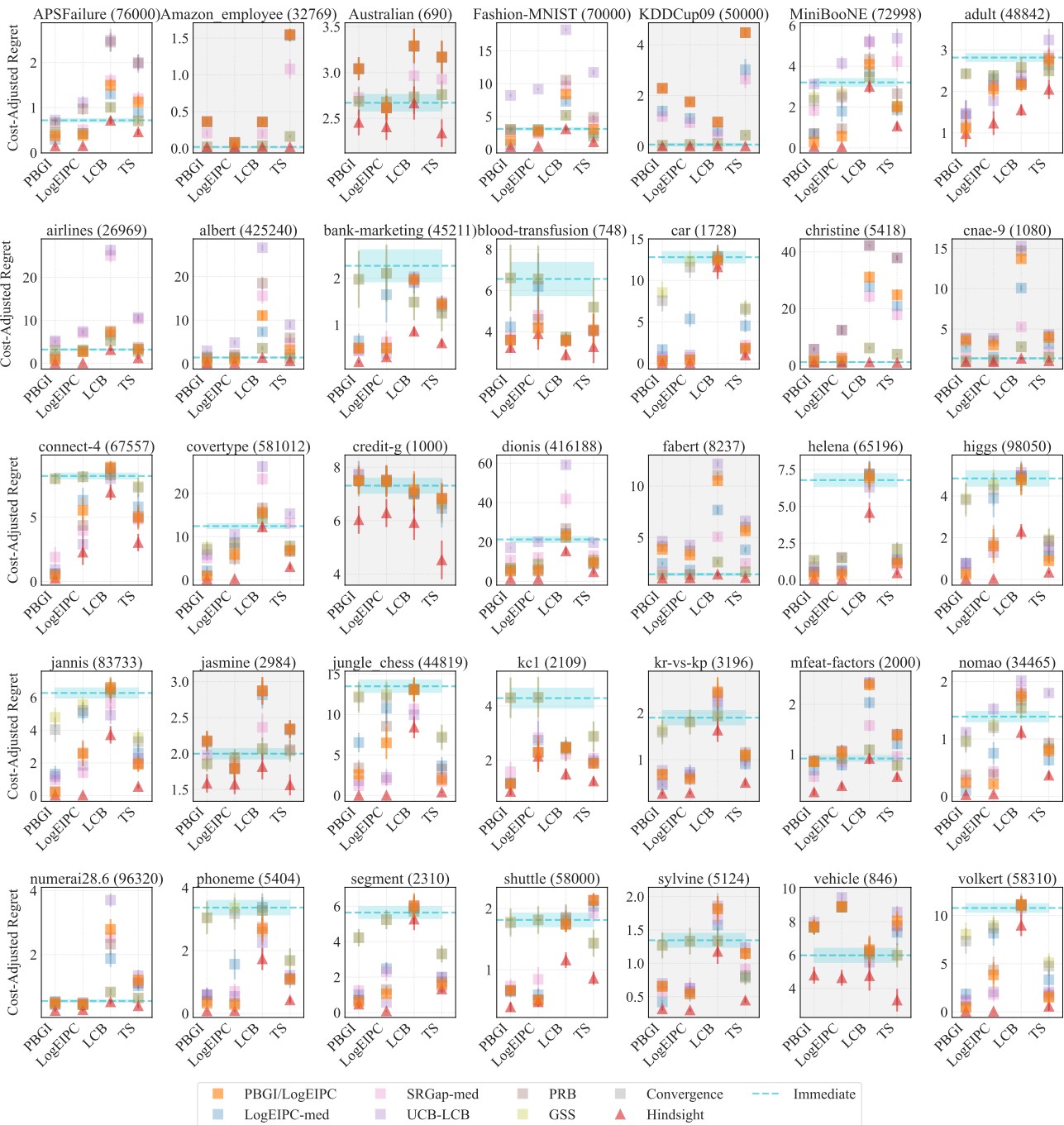

*Figure 20.* Cost-adjusted simple regret of all 35 LCBench datasets when $\lambda = 10^{-4}$. Datasets where the matched PBGI pair underperforms, meaning that it does not rank among the top three acquisition function–stopping rule pairs by cost-adjusted simple regret, are highlighted in light grey.

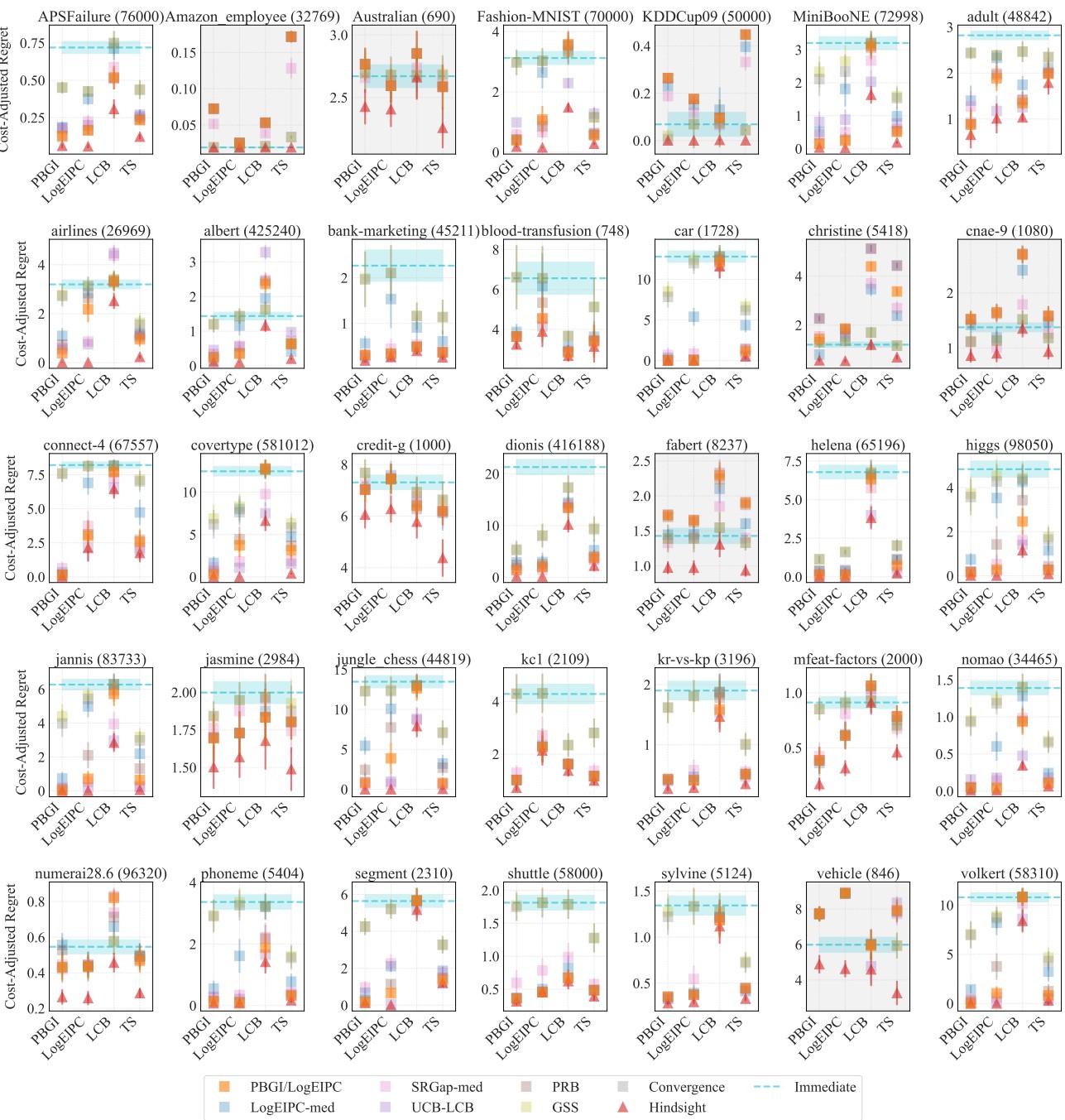

*Figure 21.* Cost-adjusted simple regret of all 35 LCBench datasets when $\lambda = 10^{-5}$. Datasets where the matched PBGI pair underperforms, meaning that it does not rank among the top three acquisition function–stopping rule pairs by cost-adjusted simple regret, are highlighted in light grey.

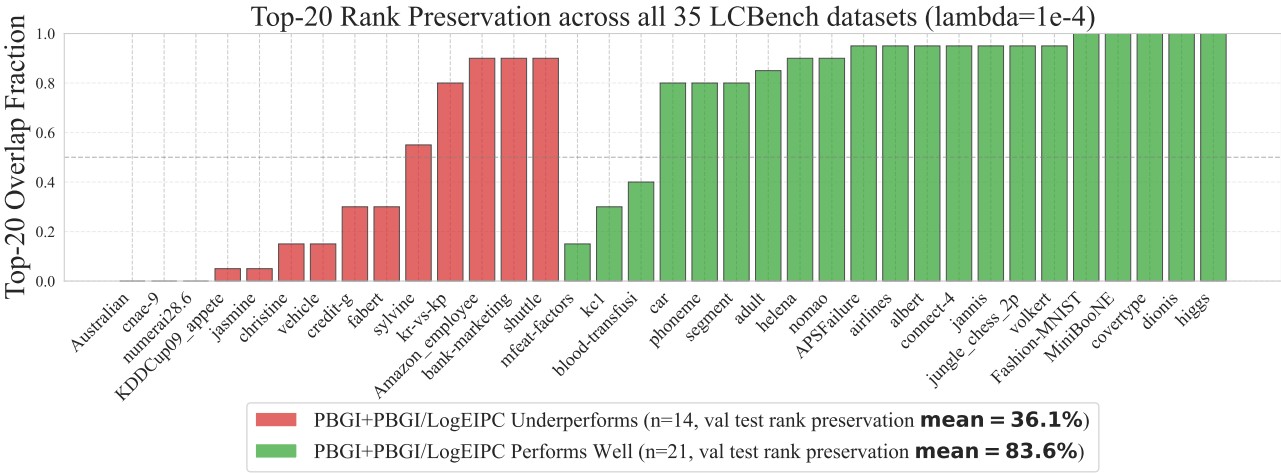

*Figure 22.* Top-20 validation–test rank preservation for every LCBench dataset. Datasets are split into $\mathcal{D}_{\mathrm{under}}$ (red) and $\mathcal{D}_{\mathrm{top}}$ (green) based on whether the matched PBGI pair ranks among the top three PBGI-based acquisition function–stopping rule pairs formed by the seven stopping rules in $\mathcal{S}$ at $\lambda = 10^{-4}$. The dashed line marks Top-20 $= 0.5$. Datasets where the matched PBGI pair underperforms also have markedly lower validation-to-test rank preservation, indicating that their weaker cost-adjusted regret reflects a val/test mismatch in the benchmark rather than a failure mode of the stopping rule itself.

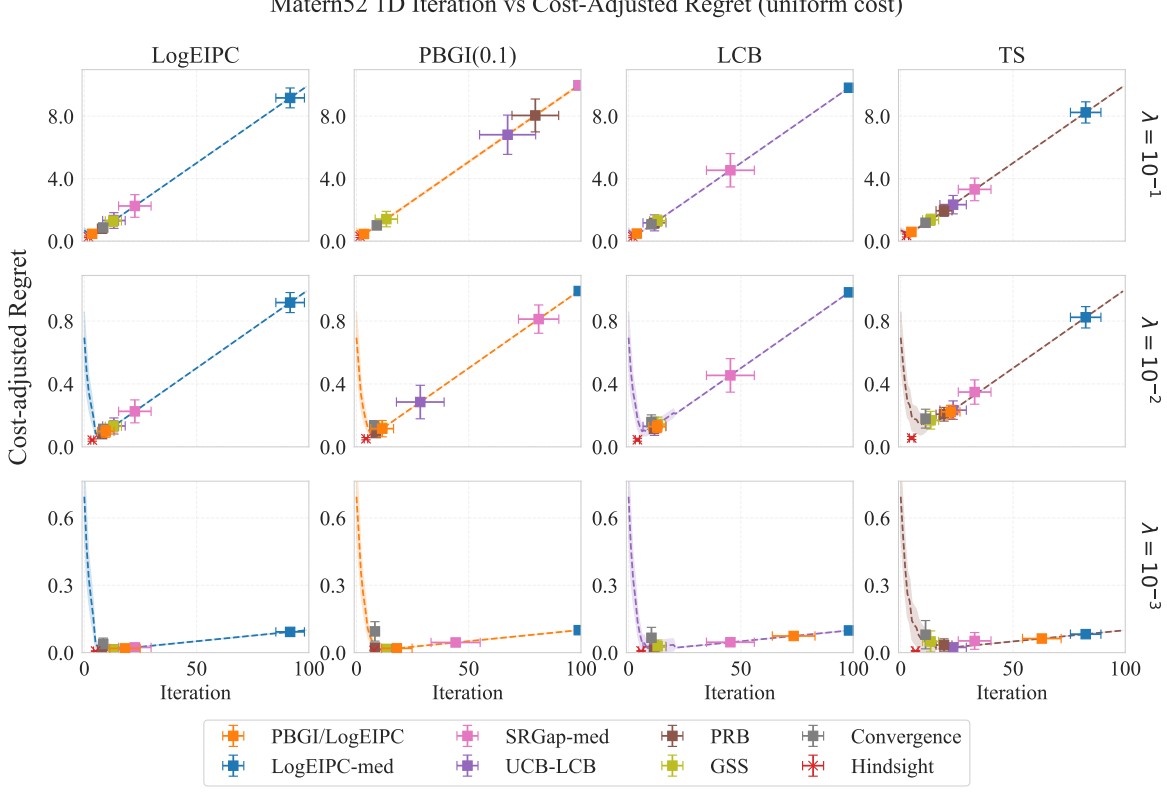

*Figure 23.* Comparison of cost-adjusted simple regret across acquisition function and stopping rule pairs in the 1D Bayesian regret experiments, with cost-scaling factor $\lambda = 10^{-1}, 10^{-2}, 10^{-3}$ and under uniform cost. The dashed line in each subplot represent fixed iteration stopping (e.g., the y-axis value of the line at iteration 50 represent the cost-adjusted regret when always stop at iteration 50).

*Figure 24.* Comparison of cost-adjusted simple regret across acquisition function and stopping rule pairs in the 1D Bayesian regret experiments, with cost-scaling factor $\lambda = 10^{-1}, 10^{-2}, 10^{-3}$ and under linear cost. The dashed line in each subplot represent fixed iteration stopping (e.g., the y-axis value of the line at iteration 50 represent the cost-adjusted regret when always stop at iteration 50).

*Figure 25.* Comparison of cost-adjusted simple regret across acquisition function and stopping rule pairs in the 1D Bayesian regret experiments, with cost-scaling factor $\lambda = 10^{-1}, 10^{-2}, 10^{-3}$ and under periodic cost. The dashed line in each subplot represent fixed iteration stopping (e.g., the y-axis value of the line at iteration 50 represent the cost-adjusted regret when always stop at iteration 50).

*Figure 26.* Comparison of cost-adjusted simple regret across acquisition function and stopping rule pairs in the 8D Bayesian regret experiments, with cost-scaling factor $\lambda = 10^{-1}, 10^{-2}, 10^{-3}$ and under uniform cost. The dashed line in each subplot represent fixed iteration stopping (e.g., the y-axis value of the line at iteration 50 represent the cost-adjusted regret when always stop at iteration 50).

*Figure 27.* Comparison of cost-adjusted simple regret across acquisition function and stopping rule pairs in the 8D Bayesian regret experiments, with cost-scaling factor $\lambda = 10^{-1}, 10^{-2}, 10^{-3}$ and under linear cost. The dashed line in each subplot represent fixed iteration stopping (e.g., the y-axis value of the line at iteration 50 represent the cost-adjusted regret when always stop at iteration 50).

*Figure 28.* Comparison of cost-adjusted simple regret across acquisition function and stopping rule pairs in the 8D Bayesian regret experiments, with cost-scaling factor $\lambda = 10^{-1}, 10^{-2}, 10^{-3}$ and under periodic cost. The dashed line in each subplot represent fixed iteration stopping (e.g., the y-axis value of the line at iteration 50 represent the cost-adjusted regret when always stop at iteration 50).

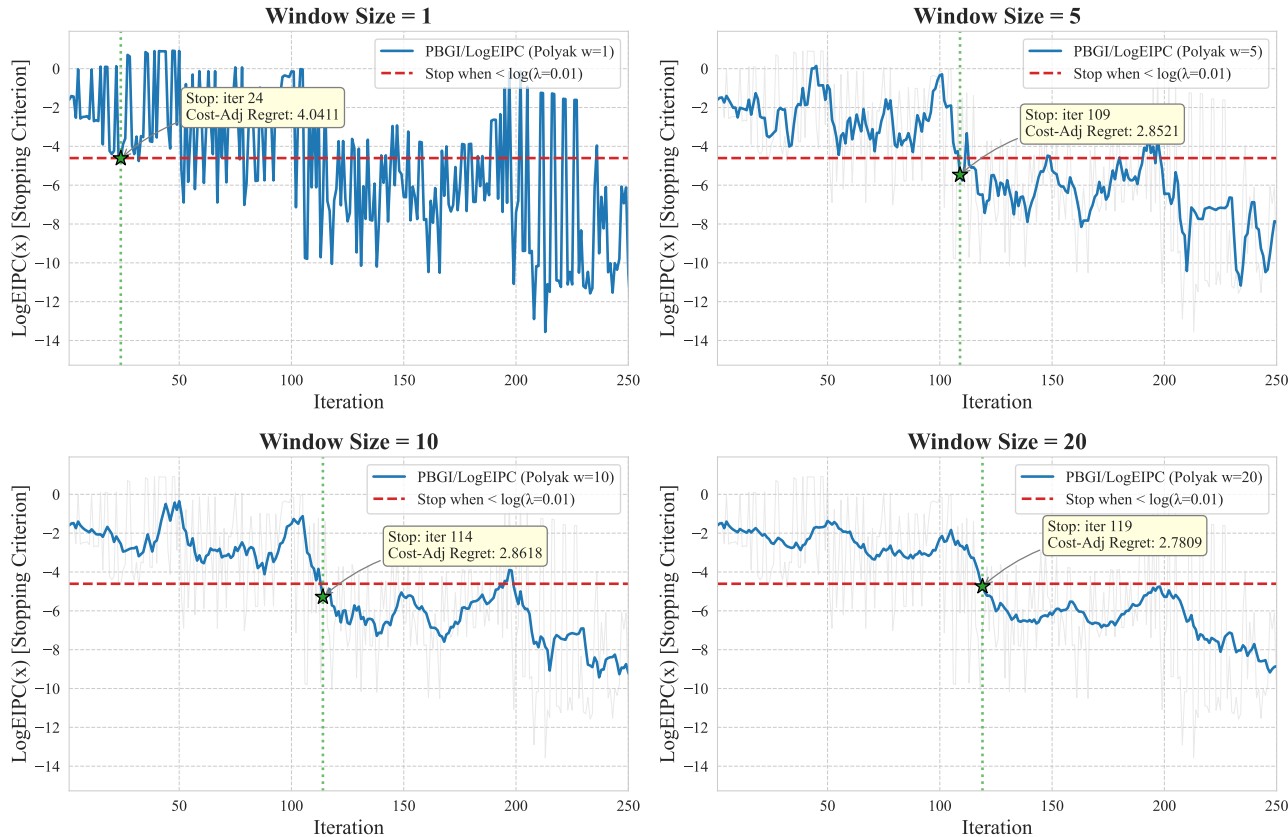

*Figure 29.* Single-trajectory illustration of how the moving-average window size $W$ affects the PBGI/LogEIPC stopping signal on the 8D Bayesian regret benchmark under periodic costs (seed 0, $\lambda = 10^{-2}$), with PBGI ($\lambda = 10^{-2}$) as the acquisition function. The light grey curve is the raw $\alpha_t^{\text{LogEIPC}}$, the blue curve its $W$-iteration trailing average, the red dashed line the stopping threshold $\log \lambda$, and the green star the resulting stopping iteration. Without smoothing ($W = 1$) the rule fires early (iter. 24, regret 4.04) on a noisy dip; smoothing defers the trigger and recovers low cost-adjusted regret (iter. 119, regret 2.78 at $W = 20$).

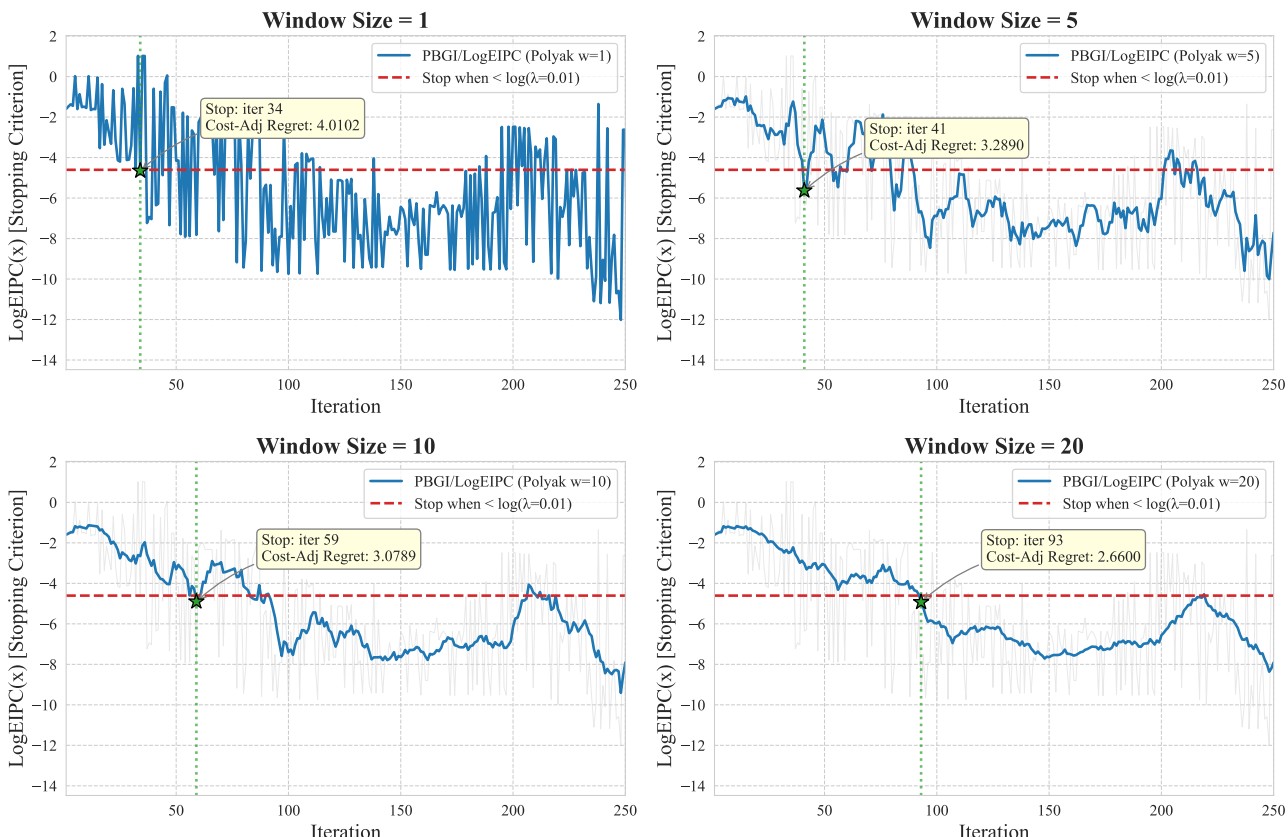

*Figure 30.* Single-trajectory illustration analogous to Figure 29, with the LogEIPC acquisition function (same seed, same cost regime, $\lambda = 10^{-2}$). Without smoothing the rule fires at iter. 34 with regret 4.01; with $W = 20$ it fires at iter. 93 with regret 2.66.

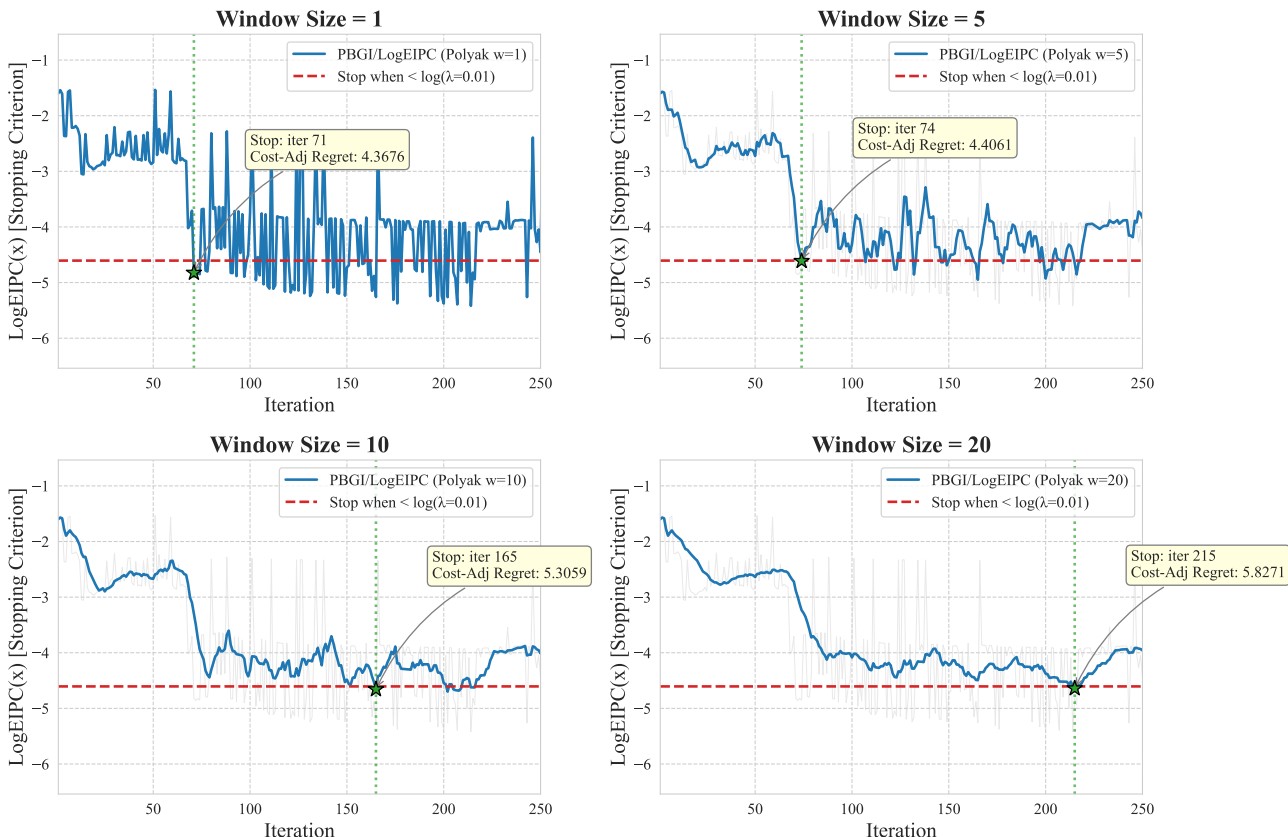

*Figure 31.* Single-trajectory illustration analogous to Figure 29, with Thompson sampling (TS) as the acquisition function. Unlike the matched-acquisition cases, smoothing here pushes the stopping iteration later (iter. 71 at $W = 1$ vs. iter. 215 at $W = 20$) without a corresponding drop in cost-adjusted regret – TS does not satisfy the expected-improvement-exceeds-cost property of Lemma 3.1, so the PBGI/LogEIPC stopping rule has no matched guarantee for it.

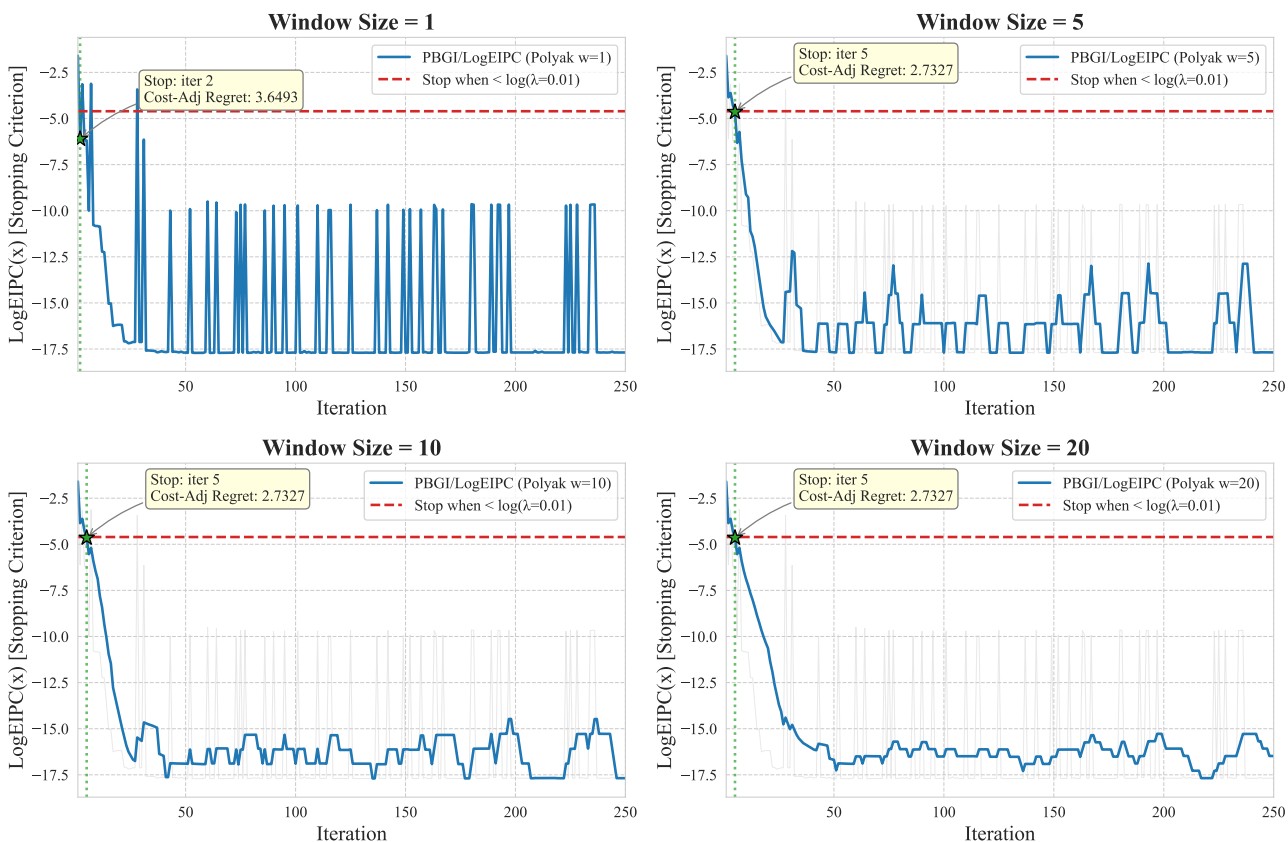

*Figure 32.* Single-trajectory illustration analogous to Figure 29, with LCB as the acquisition function. The raw LogEIPC signal has a clean transient: smoothing changes the stopping iteration only slightly (iter. 2 at $W = 1$ vs. iter. 5 at $W \geq 5$) and the cost-adjusted regret stabilises at 2.73 for all $W \geq 5$.

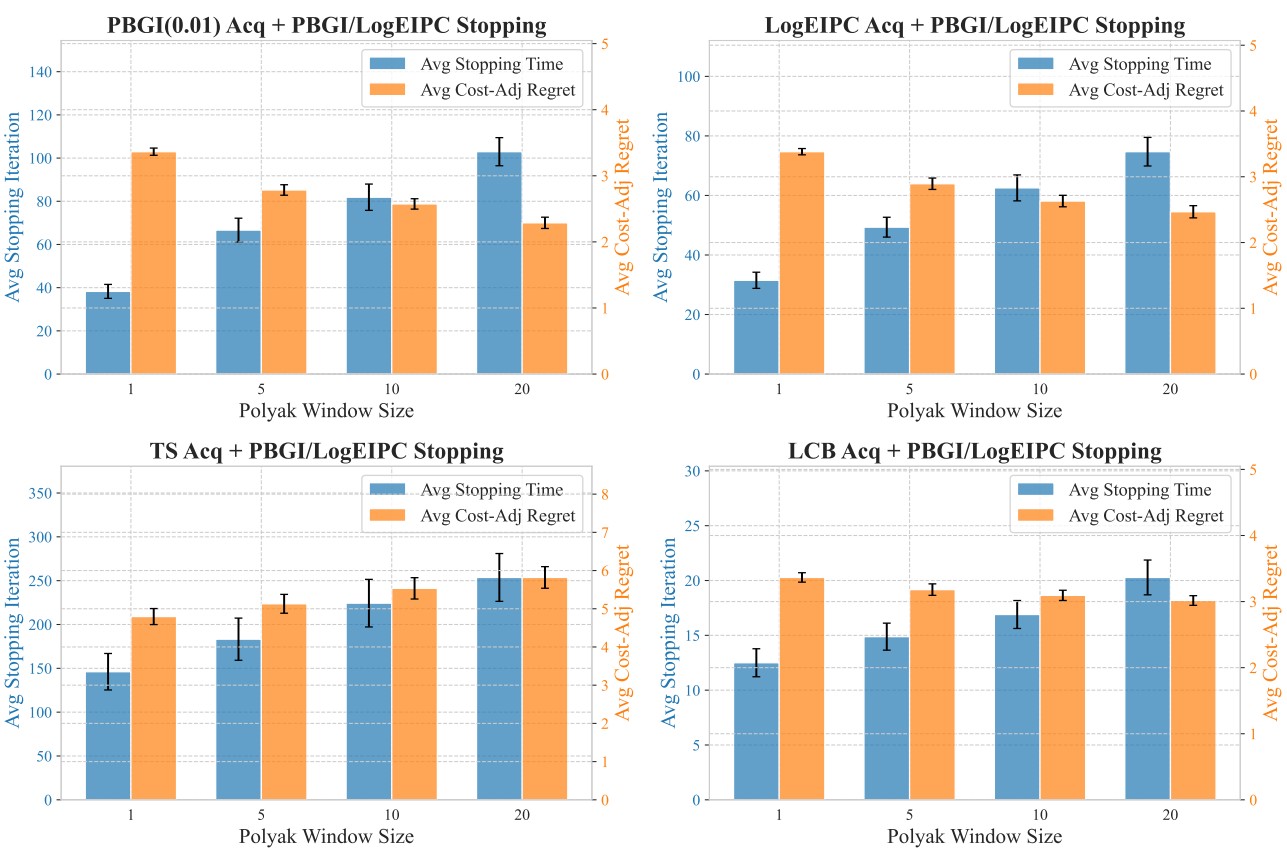

*Figure 33.* Effect of the moving-average window size $W \in \{1, 5, 10, 20\}$ on the average stopping iteration (blue, left axis) and the average cost-adjusted simple regret (orange, right axis) for the PBGI/LogEIPC stopping rule on the 8D Bayesian regret benchmark under *periodic* costs ($\lambda = 10^{-2}$, 50 seeds, error bars are $\pm 2$ standard errors). Each subplot fixes one acquisition function. Smoothing yields the largest cost-adjusted regret reductions for the matched PBGI and LogEIPC pairs, a smaller reduction for LCB, and no benefit for TS.

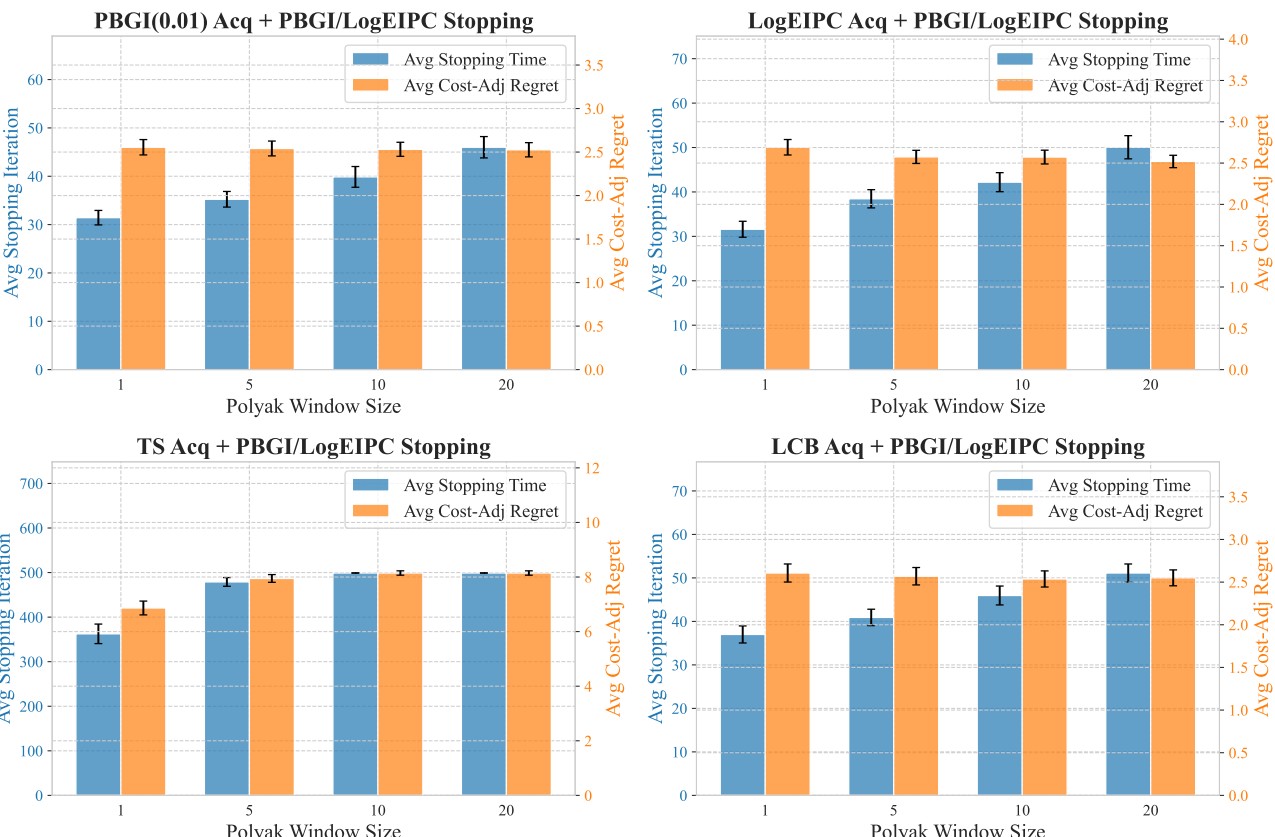

*Figure 34.* Aggregate ablation analogous to Figure 33 under *uniform* costs. Smoothing slightly reduces cost-adjusted regret for the matched PBGI and LogEIPC pairs and is roughly neutral for LCB and TS.

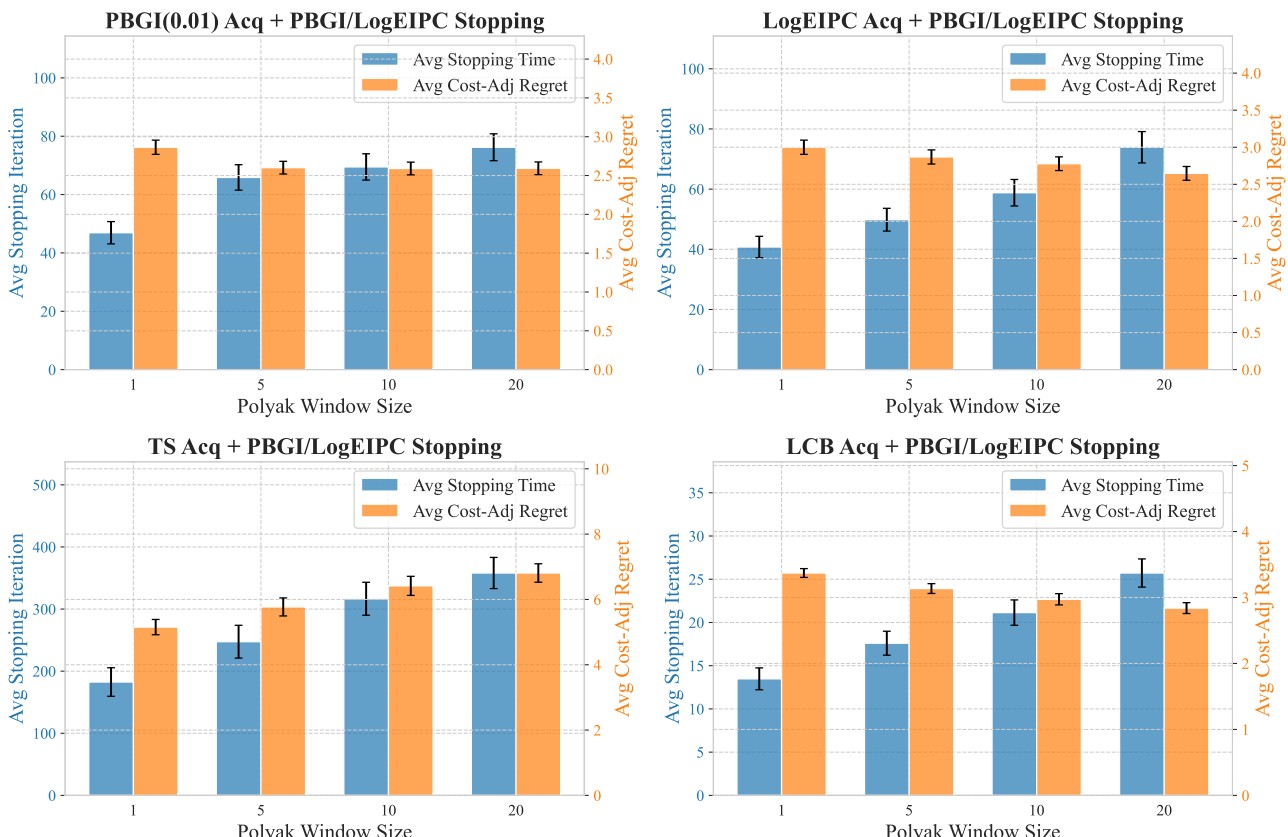

*Figure 35.* Aggregate ablation analogous to Figure 33 under *linear* costs. Smoothing reduces cost-adjusted regret for the matched PBGI and LogEIPC pairs and for LCB, with no benefit for TS.

