# OpenReview forum: "Cost-aware Stopping for Bayesian Optimization"
_ICML.cc/2026/Conference — ICML 2026 regular_

### Official Review · Reviewer_RC66 · 2026-03-06

**Soundness:** 4
**Presentation:** 3
**Significance:** 3
**Originality:** 3
**Overall Recommendation:** 5
**Confidence:** 4

**Summary:**

The paper “Cost-aware Stopping for Bayesian Optimization” aims to provide an algorithm and theoretical framework for determining when is the optimal time a Bayesian Optimization run should terminate. The authors present a cost-aware stopping rule for BO, grounded in a connection between two state-of-the-art acquisition functions: the Pandora's Box Gittins Index (PBGI) and log expected improvement per cost (LogEIPC). The authors propose the first theoretical guarantee of expected cost-adjusted regret (no worse than immediate stopping) for Bayesian Optimization, whereas previous works focused simply on heuristics for stopping criteria. Experiments are extensive. The approach also extends naturally to budgeted settings.

**Compliance With Llm Reviewing Policy:**

Affirmed.

**Final Justification:**

This is a very good paper. My initial recommendation was a clear accept and the decision was reinforced by the authors' rebuttal that addressed all the minor concerns that I had left. The work is original and very well presented. A straightforward accept decision.

**Key Questions For Authors:**

1. How significant is the guarantee of no worse than immediate stopping? Is this finding solely novel to the theoretical guarantee being derived for the first time or because other algorithms do not achieve this guarantee.
2. The debounce and stabilization modifications (Section 3.2) are critical for practical performance but break the assumptions of Theorem 3.2. Can you quantify or bound the effect of these modifications on the theoretical guarantee? Even an informal argument would help.
3. You note that ~25% of LCBench datasets show poor performance, mostly on small datasets. Have you investigated diagnostic criteria?
4. The iteration cap (100/500/200) is used as the stopping time when rules fail to trigger. How sensitive are the aggregate results? Have you tried varying it?

**Limitations:**

yes

**Strengths And Weaknesses:**

Strengths:
1. The core contribution shows that the PBGI stopping condition and the "stop when EI < cost" condition for LogEIPC are mathematically equivalent. I found it very elegant, and I appreciate how clean the derivation is. It connects two seemingly distinct lines of work (Gittins index theory via Weitzman-Kleinberg-Xie and the EI-per-cost tradition) through a monotonicity argument.
2. The presented article is technically sound. The authors provide rigorous theoretical investigations and derivations for their methods, deriving regret guarantees as well as termination criteria for their method.
3. It is very easy for the reader to follow along the narrative of the paper, very nicely written. For instance, the "stop when EI < cost" idea is clean and intuitive, easy to explain, and requires no threshold tuning beyond the natural cost-scaling factor.
4. The mixing and matching of acquisition function - stopping criteria pairs provide a very fair testing ground for their method and baselines.
5. The design of the experiments is excellent. The authors also take care to set the GP to match the true data-generating process in the Bayesian regret experiments, which isolates the stopping behavior from model misspecification, an interesting design.
6. The authors provide a large and well picked set of baseline stopping criteria to compare their model with, making it easy for the reader to believe their claims and compare the strengths and weaknesses of the proposed method. The authors also provide an easy to understand upper and lower bound in their analyses, showing where on the spectrum each algorithm lies.
7. Section 3.2 states the possible limitations and considerations of deploying this algorithm in different scenarios. This makes it clear when and when not to use this stopping criteria and how to potentially treat the weaknesses of their method.
8. The proposed solution can have significant impact in high-cost optimization problems, where each additional evaluation can incur significant costs.


Weaknesses:
1. Figures can be difficult to understand, with different scales y -axis scales making it difficult to track consistently performance between the different subfigures. Figure 1 is difficult to parse at the resolution provided; the panels are small and the legend overlaps with data.Whisker plots take too much space, cluttering the figure and requiring readers to zoom in significantly to understand. The figures throughout could benefit from a presentation pass.
2. The guarantee derived for their algorithm is that the cost-aware regret is no worse than if the algorithm were to immediately stop. In the worst case it is of course tight (all evaluations could be prohibitively expensive), but it does not tell us much about when or why the method should do well. A bound that depends on problem structure (e.g., kernel properties, cost distribution) would be considerably more informative. The authors acknowledge this is the first formal guarantee of its kind for cost-adjusted regret, which is true but could be improved.
3. Finally, each trial is capped at a fixed number of iterations (100 for 1D, 500 for 8D, 200 for empirical), and if a rule does not trigger, the cap is used as the stopping time. This cap is somewhat arbitrary, and it is not clear whether results are sensitive to this choice.

---

> ### Author Rebuttal · Authors · 2026-03-31
>
> We thank the reviewer for the detailed and encouraging feedback.
>
> ## On the significance and limitations of the theoretical guarantee
> While Theorem 3.2 may appear simple, it provides a **safeguard guarantee**. This property does not generally hold for other acquisition–stopping rule pairings, as also reflected in our empirical results. We also note that this guarantee is **tight in a worst-case sense**; we currently discuss this in the appendix and will highlight it in the main text:
>
> > Moreover, this guarantee is tight in the worst case: when evaluation costs are sufficiently large relative to the maximum achievable improvement, immediate stopping is optimal (see Proposition A.3).
>
> Beyond this, the key reason our method performs well is the **alignment between acquisition and stopping**: evaluations are taken only when expected improvement justifies their cost. This is not enforced by existing acquisition–stopping rule combinations, explaining why they may continue evaluating suboptimally.
>
>
> ## On the effect of stabilization and smoothing on the theoretical guarantees
> Our theoretical analysis assumes initialization from a single point $x_1$, whereas in practice we begin with $2(d+1)$ Sobol points followed by a stabilization period with $W=20$ points. This can be viewed as redefining the initial dataset before applying the stopping rule. The theoretical guarantees therefore extend directly by treating the end of this initialization phase as the effective starting point.
>
> We will add a remark following Theorem 3.2 and Corollary 3.3:
> > Lemma 3.1 and Theorem 3.2 are stated for simplicity under initialization from a single point $x_1$. The results extend directly to arbitrary initial datasets $(x_{1:t_0},y_{1:t_0})$ by conditioning on the observations and treating the resulting posterior as the effective prior. The same guarantees then hold for subsequent iterations $t\geq t_0$, with all quantities defined with respect to this updated posterior.
>
> Smoothing (moving average) serves a different purpose: it mitigates instability caused by imperfect acquisition optimization, especially in continuous higher-dimensional settings (see Figure 4 in Appendix B). In such cases, acquisition values may fluctuate due to failure to identify high-value points.
>
> Our guarantees characterize the **idealized setting with exact acquisition optimization**. Under imperfect optimization, the sequence of selected points may deviate, and guarantees do not directly apply. The role of smoothing is to reduce the impact of these fluctuations, thereby preventing premature stopping.
>
> We therefore view smoothing as a **practical robustness mechanism** while the theoretical guarantees describe the ideal policy.
>
>
> ## Diagnostic criteria for LCBench datasets with poor performance
> We analyze when PBGI/LogEIPC underperforms and find it coincides with **low top-k validation–test rank overlap** (i.e., fraction of configurations in the validation top-k that also appear in the test top-k). Poor performing datasets (e.g., vehicle, jasmine, cnae-9) show 0–35\% top-20 overlap, compared to 40–80\% for well-performing ones (e.g., car, phoneme, segment). See [Figure (a)](https://tinyurl.com/5y7yeu7x) for more detail.
>
> When validation rankings misalign with test rankings, improvements identified by PBGI/LogEIPC based on validation error do not transfer, leading to weaker test time performance. Consistently, on poor-performing datasets, we observe substantially better cost-adjusted regret on validation-error than on test-error ([Figure (b)](https://tinyurl.com/543ac92y)). We will include a discussion and regret comparison plots in the next version.
>
>
> ## On the iteration cap
> The iteration caps (100/500/200) are used to standardize the experimental horizon across methods for computational tractability and consistent comparison, since some stopping rules may trigger very late. They are **not part of the method**.
>
> We set the caps sufficiently large so that cost-adjusted regret has typically already reached its minimum and begins increasing thereafter (see Figures 11–16 and 22–27), with the exception of the adult dataset in Figure 12 under very small $λ$. For late-triggering rules (e.g., UCB-LCB on AutoML datasets, LogEIPC-med and SRGap-med in 1D, and PRB in 8D), increasing the cap would only lead to additional evaluations and thus higher cost-adjusted regret.
>
> A smaller cap would effectively impose a fixed-budget constraint and truncate the behavior of late-stopping rules. In contrast, our stopping rule consistently triggers well before the cap (as also shown in Table 2), so its behavior is not affected by this truncation. Our conclusions therefore focus on regimes where the cap is sufficiently large to capture the natural stopping behavior of each method.
>
>
> ## On Figure Clarity and Presentation
>
> We will revise the figures to improve readability, in particular by increasing the figure height to provide more space for each subplot.

---

> > ### Author Rebuttal · Reviewer_RC66 · 2026-03-31
> >
> > I am satisfied with the thoroughness and quality of the rebuttal. My main questions were addressed convincingly, the promised revisions (figure improvements, additional remarks on theoretical assumptions, diagnostic analysis for LCBench) will strengthen the paper. I maintain my recommendation of  5 Accept.

---

> > > ### Author Response · Authors · 2026-03-31
> > >
> > > Thank you for the positive feedback and for acknowledging our responses. We appreciate your careful review and are glad that our clarifications addressed your concerns. We will incorporate the suggested revisions to further improve the paper.

---

### Official Review · Reviewer_MDWP · 2026-03-08

**Soundness:** 3
**Presentation:** 3
**Significance:** 3
**Originality:** 2
**Overall Recommendation:** 4
**Confidence:** 4

**Summary:**

The paper proposes a simple stopping rule for Bayesian optimization when each evaluation has a cost, and the goal is to minimise regret plus total cost. This is a practically relevant setting, and an underexplored topic.
The rule is simple and straightforward, essentially to stop when the cost for evaluation is less than the expected improvement from another evaluation, although derived from Pandora's box theory.
A bound is derived, saying that the cost-adjusted regret is less than or equal to stopping immediately after the first sample.
Empirical tests on a variety of problem instances shows better performance than other approaches from the literature.

**Compliance With Llm Reviewing Policy:**

Affirmed.

**Final Justification:**

The authors have responded appropriately to my comments. I maintain my weak accept recommendation based on the low novelty and straightforward and minor theoretical results.

**Key Questions For Authors:**

1. Please provide details for engineering tricks to avoid spurious stopping.
2. Please provide an ablation study on those engineering tricks

**Limitations:**

yes

**Strengths And Weaknesses:**

Strengths:
- the paper is well written
- the topic has high practical relevance
- good empirical study
- connection between PBGI and LogEIPC

Weaknesses:
- the suggestion to stop sampling when the expected benefit is less than the cost seems almost trivial
- benefit of engineering tricks unclear and details of engineering tricks missing

Detailed comments:
The topic is practically relevant. The idea is simple and straightforward. As the authors write, others have proposed to stop when the expected improvement falls below the cost of sampling. They do typically assume uniform evaluation costs, but the extension to evaluation cost varying across the search space seems trivial and not sufficient for publication in ICML.

The most interesting part of the paper is how to prevent "spurious stops". In Section 4, additionally the use of moving averages is mentioned, probably this should be included in the "spurious stops" section. It is a pity that the impact of these engineering choices are not demonstrated in an ablation study. Also, the details of these strategies seem missing (minimum number of samples before stopping, minimum number of consistent stopping signal).

Also interesting is the comparison with the Hindsight strategy as a lower bound, indicating that the proposed stopping rule is already quite close to this lower bound, and the inclusion of the heuristic GSS strategy which seems to be surprisingly strong.

Looking at stopping rules for BO, I found one additional paper that probably should be cited: Li, S., Li, K. and Li, W., 2023. “Why Not Looking backward?” A Robust Two-Step Method to Automatically Terminate Bayesian Optimization. Advances in Neural Information Processing Systems, 36, pp.43435-43446.

It was not entirely clear to me whether in the empirical results in the main paper, the cost function was assumed known, or had to be learned. But as the Appendix has results with an explicitly unknown cost function, I assume the results in the main paper have a known cost function.

Theorem 3.2 is almost trivial, Corollary 3.3 and 3.4 are not practically useful as U is generally not known. Also, it doesn't account for the engineering tricks to avoid spurious stops, so there is a mismatch between the algorithm analysed theoretically and the algorithm evaluated empirically.

Figure 2 is difficult to read, perhaps a table would be better. Also, it would be interesting to see whether the sub-optimality comes from stopping too early or stopping too late. The appendix has some results on this aspect that could perhaps be summarised in the main paper.

For Table 1, it is not clear what the comparison algorithms are (presumably all other algorithms minus the hindsight stopping rule?).

"In every cost regime, combining our stopping rule with the PBGI, LogEIPC, or LCB acquisition function yields cost-adjusted regret that nearly matches the hindsight optimal" seems not true for LCB and lambda=0.001.

Overall, while the approach is straightforward and simple, I would still lean towards acceptance because of its practical relevance and the paper being a very nice discussion of the topic, making it a reference for researchers and practitioners.

Typos:
- Check capitalization of references (e.g., Bayesian).
- Line 378: training time scales approximately

---

> ### Author Rebuttal · Authors · 2026-03-31
>
> We thank the reviewer for the thoughtful and constructive feedback. We are glad that the reviewer finds the paper well written and practically relevant, and we especially appreciate the suggestion to better clarify the engineering details (e.g., strategies to prevent spurious stopping).
>
> ## On details of engineering tricks (spurious stopping)
> We thank the reviewer for this helpful suggestion. In the revision, we will unify the presentation of these strategies: incorporating moving averages into the "Preventing spurious stops" paragraph, removing the separate use of "debounce," and describing it directly as smoothing via a moving average. We will also clarify that smoothing applies only in the continuous setting (e.g., the 8D synthetic experiments), while the stabilization period applies more generally, and that both use the same window size $W$, yielding a single parameter. Our choice of $W=20$ matches the initial window size $I=20$ used in EI-med (Ishibashi et al., 2023).
>
> Specifically, we will revise the second part of the "Preventing spurious stops" paragraph as follows:
>
> > Second, [...] continuous higher-dimensional [...]. To handle this, we smooth the stopping signal by applying a moving average over a window of $W$ iterations, and require the stopping condition to hold consistently under this smoothed signal before stopping. See Figure 4 for an illustration. We use the same window size $W$ for both the stabilization period and smoothing to minimize the number of parameters.
>
> We will also clarify the parameter choice in the experiment section by adding:
>
> > For the stabilization and smoothing window, we use $W=20$, aligned with the initial window size $I=20$ used in EI-med (Ishibashi et al., 2023).
>
>
> ## Ablation study on moving average
> As we have noted, this smoothing trick is only necessary in higher-dimensional continuous settings. In our experiments, the issue arises exclusively in the 8D synthetic benchmark. Figure 4 illustrates the high-frequency fluctuations in the best acquisition values returned by the continuous optimizer, motivating the use of this strategy.
>
> In the ablation study, we consider smoothing window sizes of 1, 5, 10, and 20 for our PBGI/LogEIPC stopping rule. Figure [(f)](https://tinyurl.com/sxfnwjrp) [(g)](https://tinyurl.com/4hvjuzsn) [(h)](https://tinyurl.com/3vn8nec7) [(i)](https://tinyurl.com/7yzz9wcx) illustrate how the window size affects stopping and regret along a single acquisition function path.  Due to space constraints, we summarize the ablation results for PBGI/LogEIPC across all three cost settings (uniform, linear, periodic) in Figures [(c)](https://tinyurl.com/mkrmjka3) [(d)](https://tinyurl.com/54vsf285) [(e)](https://tinyurl.com/aahca8z8). The results show that applying a moving average consistently improves performance for matching acquisition functions, with the largest gains observed under periodic costs. We will include the complete ablation study in the revised version.
>
>
> ## On cause of sub-optimality in stopping
> For the LCBench datasets, suboptimal stopping is often driven by a mismatch between validation and test error, leading to unreliable early stopping signals. In the NATS benchmark, PBGI/LogEIPC can stop slightly too late, likely due to model mismatch. We will expand on this discussion in the revised version.
>
>
> ## On “triviality” of the stopping rule
> We agree that the idea is intuitive at a high level and has been considered for EI and KG under uniform evaluation costs. Our contribution is not only to extend it to varying costs, but also to justify this intuition through a principled derivation and theoretical guarantees, and address practical issues such as unknown costs and spurious stopping.
>
>
> ## Clarification on cost function assumptions in the main experiments
> We thank the reviewer for pointing this out. The main results assume a known cost function, while the unknown-cost setting is evaluated separately in Appendix C. We will make this distinction explicit in Section 4.2.
>
> ## On the practical usefulness of Corollaries 3.3–3.4 (unknown $U$)
> We agree that $U$ is generally not known exactly. However, in our AutoML experiments the objective is accuracy, so $U \le 100$ provides a simple bound. Thus, Corollaries~3.3--3.4 still yield concrete, interpretable guarantees on expected cost and stopping time without requiring exact knowledge of $U$. We will clarify this in the revision. Moreover, their role is to establish boundedness of these quantities under our stopping rule, which existing acquisition--stopping rule pairings do not guarantee.
>
>
> ## On the effect of stabilization and smoothing on the theoretical guarantees
> Please find our response to Reviewer RC66
>
>
> ## On clarifying the comparison algorithms in Table 1
> Thank you for pointing this out. The reviewer is correct: the comparison algorithms include all acquisition–stopping rule pairs except the hindsight stopping rule. We will update Table 1's caption accordingly.

---

> > ### Author Rebuttal · Reviewer_MDWP · 2026-03-31
> >
> > The authors have responded appropriately to my comments. I maintain my weak accept recommendation based on the low novelty and straightforward and minor theoretical results.

---

> > > ### Author Response · Authors · 2026-04-01
> > >
> > > Thank you for your thoughtful feedback and for taking the time to review our revisions. We are glad that our responses have addressed your concerns. We appreciate your overall assessment and your consideration of our work.

---

### Official Review · Reviewer_qzBx · 2026-03-15

**Soundness:** 3
**Presentation:** 3
**Significance:** 2
**Originality:** 2
**Overall Recommendation:** 5
**Confidence:** 4

**Summary:**

This paper proposes a stopping criterion for cost-aware Bayesian optimization (BO) where the function evaluation costs are heterogeneous. The stopping criterion is derived from Pandora’s Box Gittins index (PBGI), which is shown to be equivalent to the criterion based on the log EI per unit cost (LogEIPC) acquisition function. They provide upper bounds on the expected cost-adjusted regret when employing this stopping criterion in conjunction with either the PBGI or LogEIPC acquisition functions. Finally,  they empirically validate their approach on synthetic and tabular benchmark problems.

**Compliance With Llm Reviewing Policy:**

Affirmed.

**Key Questions For Authors:**

No major questions/concerns. One quick question – can you speculate on why the matched PBGI combination is especially strong under objective misspecification? (Line 435)

**Limitations:**

Yes

**Strengths And Weaknesses:**

Soundness – The proposed method is technically sound and supported by theoretical and experimental analyses. The crux of their methodological contribution is summarized on Page 4. The theoretical results are based on reasonable assumptions. The experiments are well-designed and quite comprehensive, benchmarking their proposed method against the state-of-the-art under a variety of conditions, such as varying dimensionalities, cost-scaling factors, and cost function families (uniform, linear, periodic). More specifically, they consider all combinations of a selection of stopping criteria with four acquisition functions. They consider synthetic problems where model misspecification is removed as a confounding factor, and AutoML benchmark problems where it is a concern. They also take care to consider the different scenarios where costs are 1) known, 2) unknown but known for performance evaluation, or 3) completely unknown.

Presentation – The manuscript is mostly clearly written and structured well. It does a fine job of contextualizing prior work and distinguishing the major categories of approaches. The overarching narrative is clearly stated and easy to follow. I believe Section 3 (the description of their core methodological contribution) could be improved, as it’s currently not very self-contained and assumes a degree of familiarity with Pandora’s box Gittins index.

Significance – This paper advances both the capabilities and practice of cost-aware Bayesian optimization by providing a way to determine when to stop the optimization procedure in a cost-efficient manner. It also improves the understanding of the LogEIPC stopping criterion by drawing connections to the proposed PBGI approach. While the scope of the impact is specific to cost-aware BO, it provides practical value and paves the way for improvements along similar lines.

Originality – The work proposes a novel and theoretically justified stopping approach that is distinct from closely related approaches. It also provides insight into and a deeper understanding of the relationship between the LogEIPC stopping criterion and the PBGI criterion, which serve as approximations of the dynamic programming problem that defines the optimal policy for cost-aware BO.

Minor comments:

* LCBench should cite Zimmer et al.

---

> ### Author Rebuttal · Authors · 2026-03-31
>
> We thank the reviewer for the positive and constructive feedback, and for recognizing the strengths of our theoretical and empirical contributions.
>
> ## LCBench citation
> Thank you for the suggestion. We note that Zimmer et al. (2021) is already cited in the first paragraph of Section 4.2 when introducing LCBench. We will add the citation to the second paragraph as well to make it more prominent.
>
>
> ## Strength of matched PBGI combination under objective misspecification
> Across our experiments (e.g., Figures 3 and 19-21), we generally observe that in empirical settings, where the surrogate model is likely misspecified to varying degrees, stopping rules based on PBGI perform better than those based on LogEIPC. We hypothesize that this might be because the PBGI acquisition function is less sensitive to model misspecification than LogEIPC when close to stopping (and the expected improvement upon best observed value is likely to be relatively small).
>
> To see this more concretely, consider a scenario where the gaussian posterior mean $\mu_t(x)$ and standard deviation $s_t(x)$ are misspecified at point $x$.
>
> For LogEIPC, $\alpha_t^{\mathrm{LogEIPC}}(x)=\log\bigl(\mathrm{EI}_t(x)/c(x)\bigr)$, where $\mathrm{EI}_t(x)$ is the posterior expected improvement at $x$. Here, the local sensitivity of $\alpha_t^{\mathrm{LogEIPC}}(x)$ to perturbations in posterior mean $\mu_t(x)$ and standard deviation $s_t(x)$ scales proportionally with  $1/\mathrm{EI}_t(x)$. When **EI is small (as is typical near stopping), even small posterior errors can induce large changes in the acquisition value**, making LogEIPC unstable under misspecification.
>
> In contrast, PBGI is defined as $\alpha_t^{\mathrm{PBGI}}(x)=g$ s.t. $\mathrm{EI}_t(x;g)=c(x)$. Thus, PBGI depends on EI only implicitly and does not involve division by EI, avoiding the amplification effect above. More concretely, its local sensitivity to posterior mean $\mu_t(x)$ is $\frac{\partial \alpha_t^{\mathrm{PBGI}}(x)}{\partial \mu_t(x)}=1$, and its local sensitivity to posterior standard deviation $s_t(x)$ is governed by a Gaussian tail ratio $\phi(z)/\Phi(z)$, where $z=\frac{g-m_t(x)}{s_t(x)}$, rather than $1/\mathrm{EI}_t(x)$. **This leads to more stable behavior when EI is small, which is precisely the regime relevant for stopping decisions.**

---

> > ### Author Rebuttal · Reviewer_qzBx · 2026-04-01
> >
> > N/A

---

> > > ### Author Response · Authors · 2026-04-01
> > >
> > > Thank you for your feedback and the insightful question.

---

### Official Review · Reviewer_zaiS · 2026-03-15

**Soundness:** 4
**Presentation:** 3
**Significance:** 3
**Originality:** 2
**Overall Recommendation:** 4
**Confidence:** 4

**Summary:**

The submission looks at cost sensitive stopping rules for Bayesian optimization.  It makes a connection between the Pandora’s
Box Gittins Index (PBGI) and log expected improvement per cost (LogEIPC) in Section 3.  Eq (8) shows that thresholding LogEIPC at zero is a sensible choice of stopping criterion as it stops when the expected improvement is below the cost of evaluating the next point.  A regret analysis is performed in Section 3.1, and some benchmark results are shown in 4.1 and 4.2.

**Compliance With Llm Reviewing Policy:**

Affirmed.

**Final Justification:**

The authors have addressed my concerns about relationship to previous work.

The authors perhaps incorrectly state that (log)EIPC and EIC are different acquisition functions: $\arg\max_x EI(x)/c(x) = \arg\max_x \log(EI(x)) - \log(c(x))$. The logEIPC can now be interpreted as a Lagrangian of a constrained optimization where $\lambda=1$: $\arg\max_x \log(EI(x)) - \lambda \log(c(x)) = \arg\max_x \log(EI(x)) \text{s.t.} \log(c(x)) \leq B$.  The constraint can be rewritten as $\exp\log(c(x)) = c(x) \leq \exp(B)$, which is just another non-negative constant.  Maximizing the monotonic transformation $\exp(\log(EI(x)))$ and taking a Lagrangian gives the EIC acquisition.  The units of the Lagrange multiplier transform $c(x)$ into the units of $EI(x)$, e.g. USD.  Appropriately setting this Lagrange multiplier and absorbing it into the definition of $c(x)$ (the units of a linear combination must be equivalent) shows that any instantiation of (log)EIPC has an equivalent EIC with the same maximizer.

**Key Questions For Authors:**

Thresholding the optimized acquisition function is a common termination criterion, including for regret analysis.  It is equivalent to the LogEIPC objective in the case of fixed costs.  Could this discussion (and reference above) be added to the text?

**Limitations:**

Yes.

**Strengths And Weaknesses:**

Considering stopping criteria for Bayesian optimization that are cost sensitive is a good thing to do.  The results seem to be well derived, and the text is presented fairly clearly, though a clearer explicit formulation of PBGI early in the paper would be good.  Stopping criteria for Bayesian optimization has often been somewhat neglected in the literature.  It appears that the Expected Improvement stopping criterion presented in Eq (8) is analyzed in Sec. 4.2 of [1] in the case of constant cost per evaluation, and is connected to expected profit maximization as well as regret.

[1] Zhou et al., A Corrected Expected Improvement Acquisition Function Under Noisy Observations. ACML 2023.

---

> ### Author Rebuttal · Authors · 2026-03-31
>
> ## Connection to existing EI thresholding-based stopping rules
> Thank you for the helpful suggestion and for pointing us to this reference.
>
> We would like to clarify that we already state in the Introduction (page 2) that our stopping rule reduces to an existing EI thresholding rule in the case of constant evaluation cost, and we cite Nguyen et al. (2017) in this context. We will strengthen this connection by adding a citation to Zhou et al. (2023) and clarifying in both the Introduction and the literature review (Section 2.2) that some prior work (Nguyen et al., 2017; Zhou et al., 2023, Frazier \& Powell, 2008) considers stopping when the EI or KG acquisition value falls below the cost per sample.
>
> We will also add the following sentence after Eq. (9) in Section 3:
>
> > When $c(x)\equiv c_0$, Equation (8) reduces to the EI thresholding stopping rule (Nguyen et al., 2017; Zhou et al., 2023)
> $$\max_{x\in X\backslash \lbrace x_1, \dots, x_t \rbrace} \alpha_t^{\mathrm{EI}}(x; y^*_{1:t}) \leq c_0.$$
>
> We would also like to clarify that while stopping when expected improvement falls below the evaluation cost is a natural idea, our contribution is not simply to adopt this rule, but to derive it from a **principled perspective grounded in Gittins index theory**, and to identify the conditions under which the PBGI/LogEIPC stopping rule performs well. Specifically, our stopping rule is most effective when paired with acquisition functions that share its underlying principles (PBGI and LogEIPC): if there exists a point where the expected improvement exceeds the cost, these acquisition functions are guaranteed to select one. Our empirical results (Figures 2–3) corroborate this, showing substantially better performance of the PBGI/LogEIPC stopping rule when paired with a matching acquisition function (PBGI, LogEIPC) compared to acquisition functions based on different principles (LCB, TS). Furthermore, our formulation is specifically designed for the more general setting of **varying and unknown evaluation costs**, a practically important regime that has received limited attention in existing stopping rules.
>
>
> ## Presentation improvements
> We will also revise the presentation of the PBGI acquisition function to improve clarity. Specifically, we will make its interpretation more explicit by changing
>
> >where the value $g$ is the threshold at which the expected improvement equals the cost
>
> to
>
> >where the value $g$ is the threshold such that the expected improvement over $g$ equals the evaluation cost
>
> clarifying that the expected improvement is taken with respect to the threshold $g$.

---

> > ### Author Rebuttal · Reviewer_zaiS · 2026-03-31
> >
> > The rebuttal addresses my main concern about relation to prior work and clarification about the claimed delta.  The EIPC (I am less convinced of the importance of distinguishing LogEIPC as it's just a monotonic transformation) is a sensible acquisition to threshold, and it is quite natural to do so when the denominator has an appropriate interpretation as a cost relative to the expected improvement.  A linear combination of negative evaluation cost and expected improvement would also be equivalent, and have a very straightforward interpretation of an eventual weighting parameter being a Lagrange multiplier transforming the units of the objective and cost function to be the same - this weighting would not be necessary if the cost function would be written in the correct units.  Evaluating this in the context of variable evaluation cost is a good thing to do, provided such a cost function can be formulated.

---

> > > ### Author Response · Authors · 2026-04-01
> > >
> > > Thank you for this insightful point! We agree that a linear combination of expected improvement and cost provides a natural alternative perspective. We believe this corresponds to the expected improvement minus cost (EI-C) acquisition function (Hu et al., 2025). As noted in the appendix, we show that EI-C admits an equivalent stopping rule (though it differs as an acquisition function), and that this stopping rule also satisfies our theoretical guarantees.
> > >
> > > [1] Hu, S., Wang, H., Dai, Z., Low, B.K.H. and Ng, S.H., 2025. Adjusted expected improvement for cumulative regret minimization in noisy bayesian optimization. Journal of Machine Learning Research, 26(46), pp.1-33.
> > >
> > > ## Clarification On Final Justification
> > >
> > > We briefly clarify the relationship between (log)EIPC and EI–C.
> > >
> > > We understand the reviewer’s argument as suggesting that, for a fixed posterior at time $t$, one can choose a value of $\lambda$ so that maximizing $EI_t(x) - \lambda  c(x)$ yields the same maximizer as maximizing $EI_t(x) / c(x)$. This statement is correct, but it is specific to a single timestep.
> > >
> > > Indeed, let $x_{t,\text{opt}}$ be a maximizer of $EI_t(x) / c(x)$, and define  $\lambda_t = EI_t(x_{t,\text{opt}}) / c(x_{t,\text{opt}})$.
> > > Then for all $x$, $EI_t(x) - \lambda_t  c(x) \le 0$, with equality at $x_{t,\text{opt}}$.
> > > Therefore, $x_{t,\text{opt}}$ also maximizes $EI_t(x) - \lambda_t  c(x)$.
> > >
> > > However, this does not imply that there exists a fixed $\lambda$ such that $EI_t(x) - \lambda  c(x)$ is equivalent to (log)EIPC across timesteps. The function $EI_t(\cdot)$ changes after each Bayesian update, and so the corresponding value of $\lambda_t$ also changes.
> > >
> > > To make this concrete, consider a domain $\{a,b,c\}$ with costs $c(a)=3$, $c(b)=1$, and $c(c)=2$. At timestep 1, suppose
> > > $EI_1(a)=1.807$, $EI_1(b)=0.422$, and $EI_1(c)=1.150$.
> > >
> > > Then (log)EIPC selects $a$, since $1.807/3=0.602$ exceeds $0.422$ and $1.150/2=0.575$. For EI–C to select the same point, we require
> > > $1.807 - 3 \lambda \ge 0.422 - \lambda$, and
> > > $1.807 - 3 \lambda \ge 1.150 - 2 \lambda$,
> > > which implies $\lambda \le 0.657$.
> > >
> > > Now suppose $a$ is evaluated and the posterior is updated, yielding
> > > $EI_2(a)=0$, $EI_2(b)=3.595$, and $EI_2(c)=4.715$.
> > >
> > > Then (log)EIPC selects $b$, since $3.595 > 4.715/2 = 2.358$. For EI–C to select $b$, we require
> > > $3.595 - \lambda \ge 4.715 - 2 \lambda$,
> > > i.e., $\lambda \ge 1.121$. These conditions are incompatible, so no fixed $\lambda$ can reproduce the same sequence of maximizers across timesteps.
> > >
> > > More generally, this equivalence does not extend to a fixed schedule $\{\lambda_t\}$ chosen a priori (e.g., analogous to the $\beta_t$ schedule in GP-UCB). The value $\lambda_t$ depends on the current EI function, and in particular on the maximizer of $EI_t(x)/c(x)$ itself, so it is inherently data-dependent and cannot be specified in advance.

---

### Decision · Program_Chairs · 2026-04-30

**Decision:**

Accept (regular)

**Comment:**

This papers presents a cost-aware stopping rule for BO leveraging Pandora's Box Gittins Index (PBGI) and log expected improvement per cost (LogEIPC). The paper was found to be well written and executed, with extensive numerical tests and some rather straightfoward but original theoretical result on an understudied topic of great practical relevance. The discussion was constructive and is expected to allow further improving the paper towards publication in ICML 2026.